# HiT-MDP: Learning the SMDP option framework on MDPs with Hidden Temporal Embeddings

**Chang Li**[1],    **Dongjin Song**[2],    **Dacheng Tao**[1]

[1]The University of Sydney,    [2]University of Connecticut

spacegoing@gmail.com, dongjin.song@uconn.edu, dacheng.tao@sydney.edu.au

## Abstract

The standard option framework is developed on the Semi-Markov Decision Process (SMDP) which is unstable to optimize and sample inefficient. To this end, we propose the Hidden Temporal MDP (HiT-MDP) and prove that the option-induced HiT-MDP is homomorphic equivalent to the option-induced SMDP. A novel transformer-based framework is introduced to learn options' embedding vectors (rather than conventional option tuples) on HiT-MDPs. We then derive a stable and sample efficient option discovering method under the maximum-entropy policy gradient framework. Extensive experiments on challenging *Mujoco* environments demonstrate HiT-MDP's efficiency and effectiveness: under widely used configurations, HiT-MDP achieves competitive, if not better, performance compared to the state-of-the-art baselines on all finite horizon and transfer learning environments. Moreover, HiT-MDP significantly outperforms all baselines on infinite horizon environments while exhibiting smaller variance, faster convergence, and better interpretability. Our work potentially sheds light on the theoretical ground of extending the option framework into a large scale foundation model.

## 1 Introduction

The option framework (Sutton et al., 1999) is one of the most promising frameworks to enable RL methods to conduct lifelong learning (Mankowitz et al., 2016) and has proven benefits in speeding learning (Bacon, 2018), improving exploration (Harb et al., 2018), and facilitating transfer learning (Zhang & Whiteson, 2019). Standard option framework is developed on Semi-Markov Decision Process, we refer to this as SMDP-Option. In SMDP-Option, an option is a temporally abstracted action whose execution cross a variable amount of time steps. A master policy is employed to compose these options and determine which option should be executed and stopped.

The SMDP formulation has two deficiencies that severely impair options' applicability in a broader context (Jong et al., 2008). The first deficiency is sample inefficiency. Since the execution of an option persists over multiple time steps, one update of the master policy consumes various steps of samples and thus is sample inefficient (Levy & Shimkin, 2011; Daniel et al., 2016; Bacon et al., 2017). The second deficiency is unstable optimizing algorithms. SMDP-based optimization algorithms are notoriously sensitive to hyperparameters. Therefore, they often exhibit large variance (Wulfmeier et al., 2020) and encounter convergence issues (Klissarov et al., 2017).

Extensive research has tried to tackle these issues from aspects such as improving the policy iteration procedure (Sutton et al., 1999; Daniel et al., 2016; Bacon et al., 2017) and adding extra constraints on option discovering objectives (Khetarpal et al., 2020; Wulfmeier et al., 2020; Hyun et al., 2019). However, rare works (Levy & Shimkin, 2011; Smith et al., 2018; Zhang & Whiteson, 2019) explore from the perspective of improving the underlying Decision Process. Our work is largely different from literatures above and more details are discussed in Section 6. In this work, we present a counterintuitive finding: the SMDP formulated option framework has an MDP equivalence which is still able to temporally extending the execution of abstracted actions.

MDP-based options can address SMDP-based ones' deficiencies from two aspects: (1) sample efficient, *i.e.*, MDPs policies can be optimized at every sampling step (Bacon, 2018); and (2) more stable to optimize, *i.e.*, convergence of MDPs algorithms are well theoretically justified and have smaller variance (Schulman et al., 2015). In this paper, we propose the Hidden Temporal MDP (HiT-MDP) and theoretically prove the equivalence to the SMDP-Option. We first formulate HiT-

MDP as an HMM-like PGM and introduce temporally dependent latent variables into the MDP to preserve temporal abstractions. By exploiting conditional independencies in PGM, we prove that the HiT-MDP is *homomorphic equivalent* (Ravindran, 2003) to SMDP-Option. To the best of our knowledge, this is the first work proposing an MDP equivalence of the standard option framework.

In order to solve optimal values of the HiT-MDPs, we devise a *Markovian Option-Value Function* $\bar{V}[\mathbf{s}_t, \mathbf{o}_{t-1}]$ and prove that it is an unbiased estimation of the standard value function $V[\mathbf{s}_t]$. We further develop the Hidden Temporal Bellman Equation in order to derive the policy evaluation theorem for HiT-MDPs. We also show that the *Markovian Option-Value Function* has a variance reduction effect. As a result, HiT-MDP is a general-purpose MDP that can be updated at every sampling step, and thus naturally address the sample inefficiency issue.

We solve the learning problem by deriving a stable on-policy policy gradient method under the maximum entropy reinforcement learning framework. One difficulty of learning standard option frameworks is that they do not have any constraint on qualities of options (Harb et al., 2018). Standard option frameworks have tendencies to learn either degenerate options (Harb et al., 2018) (short execution time) that switching back-and-forth frequently, or dominant options (Zhang & Whiteson, 2019) (long execution time) that executing through the whole episode. We tackle this problem by proposing the Maximum entropy Options Policy Gradient (MOPG) algorithm. MOPG includes an information-theoretical intrinsic reward to encourage consecutive executions of options and entropy terms to encourage explorations of options. The whole algorithm can be solved in an end-to-end manner under the structional variational inference framework. We theoretically prove that optimizing through MOPG converges to the optimal trajectory.

We conduct experiments on challenging *Mujoco* (Todorov et al., 2012; Brockman et al., 2016b; Tunyasuvunakool et al., 2020) environments. Thorough empirical results demonstrate that under widely used configurations, HiT-MDP achieves competitive, if not better, performance compared to the state-of-the-art baselines on all finite horizon and transfer learning environments. Moreover, HiT-MDP significantly outperforms all baselines on infinite horizon environments while exhibiting smaller variance, faster convergence, and interpretability.

## 2 BACKGROUND

**Markov Decision Process:** A Markov Decision Process (Puterman, 1994) $M = \{\mathbb{S}, \mathbb{A}, r, P, \gamma\}$ consists of a state space $\mathbb{S}$, an action space $\mathbb{A}$, a state transition function $P(\mathbf{s}_{t+1}|\mathbf{s}_t, \mathbf{a}_t) : \mathbb{S} \times \mathbb{A} \to \mathbb{S}$, a discount factor $\gamma \in \mathbb{R}$, and a reward function $r(\mathbf{s}, \mathbf{a}) = \mathbb{E}[r|\mathbf{s}, \mathbf{a}] : \mathbb{S} \times \mathbb{A} \to \mathbb{R}$ which is the expectation of the reward $r_{t+1} \in \mathbb{R}$ received from the environment after executing action $\mathbf{a}_t$ at state $\mathbf{s}_t$. A policy $\pi = P(\mathbf{a}|\mathbf{s}) : \mathbb{A} \times \mathbb{S} \to [0, 1]$ is a probability distribution defined over actions conditioning on states. A discounted return is defined as $G_t = \sum_k^N \gamma^k r_{t+k+1}$, where $\gamma \in (0, 1)$ is a discounting factor. The value function $V[\mathbf{s}_t] = \mathbb{E}_{\tau \sim \pi}[G_t|\mathbf{s}_t]$ is the expected return starting at state $\mathbf{s}_t$ and the trajectory $\tau = \{\mathbf{s}_t, \mathbf{a}_t, r_{t+1}, \mathbf{s}_{t+1}, \dots\}$ follows policy $\pi$ thereafter. The action-value function is defined as $Q[\mathbf{s}_t, \mathbf{a}_t] = \mathbb{E}_{\tau \sim \pi}[G_t|\mathbf{s}_t, \mathbf{a}_t]$.

**Homomorphic Equivalence:** Givan et al. (2003) define the equivalence relation between MDPs as symmetric equivalence (bisimulation relation). Ravindran (2003) extends their work to *homomorphic equivalence* that allows defining symmetries between an MDP and SMDP. Given two processes, an MDP $M = \{\mathbb{S}, \mathbb{A}, R, P, \gamma\}$ with the trajectory $\tau$ and an SMDP $\tilde{M} = \{\tilde{\mathbb{S}}, \mathbb{A}, \tilde{R}, \tilde{P}, \tilde{\gamma}\}$ with the trajectory $\tilde{\tau}$. Assume both $M$ and $\tilde{M}$ share the same action space $\mathbb{A}$. An *homomorphism* $\tilde{B}$ is a tuple of surjection partition functions, $M$ and $\tilde{M}$ is *Homomorphic Equivalence* if 1) for all state-action pairs $\{\mathbf{s}, \mathbf{a}\}$, there exists a *many-to-one correspondence* equivalent state-action pair $\{\tilde{\mathbf{s}}, \tilde{\mathbf{a}}\}$ that $\{\mathbf{s}, \mathbf{a}\}/\tilde{B} = \{\tilde{\mathbf{s}}, \tilde{\mathbf{a}}\}/\tilde{B}$, or denoted as $\tilde{B}(\{\tilde{\mathbf{s}}, \tilde{\mathbf{a}}\}) = \{\mathbf{s}, \mathbf{a}\}$, 2) and following conditions hold:

1.          $P(\tau/\tilde{B}) \equiv P(\tilde{\tau}/\tilde{B}), \quad$ and $\tilde{B}$ is a *surjection*,
2.          $r(\tau/\tilde{B}) \equiv r(\tilde{\tau}/\tilde{B}),$

**The SMDP-based Option Framework**: In *SMDP-Option* (Sutton et al., 1999; Bacon, 2018), an option is a triple $(\mathbb{I}_o, \pi_o, \beta_o) \in \mathcal{O}$, where $\mathcal{O}$ denotes the option set; the subscript $o \in \mathbb{O} = \{1, 2, \dots, K\}$ is a positive integer index which denotes the $o$-th triple where $K$ is the number of options; $\mathbb{I}_o$ is an initiation set indicating where the option can be initiated; $\pi_o = P_o(\mathbf{a}|\mathbf{s}) : \mathbb{A} \times \mathbb{S} \to [0, 1]$ is the action policy of the $o$th option; $\beta_o = P_o(\mathbf{b} = 1|\mathbf{s}) : \mathbb{S} \to [0, 1]$ where $\mathbf{b} \in \{0, 1\}$ is a *termination function*. For clarity, we use $P_o(\mathbf{b} = 1|\mathbf{s})$ instead of $\beta_o$ which is widely used in previous

option literatures (e.g., Sutton et al. (1999); Bacon et al. (2017)). A *master policy* $\pi(\mathbf{o}|\mathbf{s}) = P(\mathbf{o}|\mathbf{s})$ where $\mathbf{o} \in \mathbb{O}$ is used to sample which option will be executed. Therefore, the dynamics (stochastic process) of the option framework is written as:

$$P(\tau) = P(\mathbf{s}_0)P(\mathbf{o}_0)P_{o_0}(\mathbf{a}_0|\mathbf{s}_0) \prod_{t=1}^{\infty} P(\mathbf{s}_t|\mathbf{s}_{t-1}, \mathbf{a}_{t-1})P_{o_t}(\mathbf{a}_t|\mathbf{s}_t)$$

$$[P_{o_{t-1}}(\mathbf{b}_t = 0|\mathbf{s}_t)\mathbf{1}_{\mathbf{o}_t=o_{t-1}} + P_{o_{t-1}}(\mathbf{b}_t = 1|\mathbf{s}_t)P(\mathbf{o}_t|\mathbf{s}_t)], \tag{1}$$

where $\tau = \{\mathbf{s}_0, \mathbf{o}_0, \mathbf{a}_0, \mathbf{s}_1, \mathbf{o}_1, \mathbf{a}_1, \ldots\}$ denotes the trajectory of the option framework. $\mathbf{1}$ is an indicator function and is only true when $\mathbf{o}_t = o_{t-1}$ (notice that $o_{t-1}$ is the realization at $\mathbf{o}_{t-1}$). Therefore, under this formulation the option framework is defined as a Semi-Markov process since the dependency on an activated option $o$ can cross a variable amount of time (Sutton et al., 1999).

## 3 AN MDP EQUIVALENCE OF THE SMDP-BASED OPTION FRAMEWORK

In this section, we propose the Hidden Temporal MDP (HiT-MDP). We first reformulate the *SMDP-Option* as an HMM-like Probabilistic Graphical Model (PGM). We then propose the HiT-MDP as a marginalization of the PGM and prove that the HiT-MDP is *homomorphic equivalent* (Ravindran, 2003) to the *SMDP-Option*. We also derive the HiT-MDP's Bellman Equation and prove its convergence by presenting the policy evaluation theorem. As a result, HiT-MDP is a general-purpose MDP that can be combined with any policy optimization algorithm off-the-shelf. We propose an efficient learning algorithm and complete the proof of the policy iteration theorem in Section 4.

### 3.1 AN MDP FORMULATION OF THE OPTION FRAMEWORK

Following Bishop (2006)'s formulation of mixture distributions, we redefine the option random variable $\mathbf{o} \in \mathbb{O} = \{1, 2, \ldots, K\}$, which was originally defined as an integer index, but now as a $K$-dimensional one-hot vector $\bar{\mathbf{o}} \in \bar{\mathbb{O}} = \{0, 1\}^K$ where $K$ is the number of options. We further employ the one-hot vector to reformulate the *termination function* and action function of each option into two mixture distributions by introducing extra dependencies on $\bar{\mathbf{o}}$:

$$P(\mathbf{a}_t|\mathbf{s}_t, \bar{\mathbf{o}}_t) = \prod_{o \in \bar{\mathbf{o}}_t} P_o(\mathbf{a}_t|\mathbf{s}_t)^o, \qquad P(\mathbf{b}_t|\mathbf{s}_t, \bar{\mathbf{o}}_{t-1}) = \prod_{o \in \bar{\mathbf{o}}_{t-1}} P_o(\mathbf{b}_t|\mathbf{s}_t)^o \tag{2}$$

Since the option random variable $\bar{\mathbf{o}}$ is now a one-hot vector, an instantiation $\bar{o} \triangleq k$ denotes the activation of the option $k$, and by definition only the $k$-th entry of $\bar{o}_t$ is 1 and all the other entries are 0. Therefore, we have $P_{o_t}(\mathbf{a}_t|\mathbf{s}_t) = P(\mathbf{a}_t|\mathbf{s}_t, \bar{\mathbf{o}} = \bar{o}_t)$ and $\beta_{o_{t-1}} = P_{o_{t-1}}(\mathbf{b}_t = 1|\mathbf{s}_t) = P(\mathbf{b}_t = 1|\mathbf{s}_t, \bar{\mathbf{o}}_{t-1} = \bar{o}_{t-1})$.

The third reformulation is that we propose a novel *MDP mixture master policy* $P(\bar{\mathbf{o}}_t|\mathbf{s}_t, \mathbf{b}_t, \bar{\mathbf{o}}_{t-1})$, which is a mixture distribution containing the *SMDP master policy* $P(\bar{\mathbf{o}}_t|\mathbf{s}_t)$ and a degenerate probability as mixture components by adding two extra dependencies on $\mathbf{b}_t$ and $\bar{\mathbf{o}}_{t-1}$:

$$P(\bar{\mathbf{o}}_t|\mathbf{s}_t, \mathbf{b}_t, \bar{\mathbf{o}}_{t-1}) = P(\bar{\mathbf{o}}_t|\mathbf{s}_t)^{\mathbf{b}_t} P(\bar{\mathbf{o}}_t|\bar{\mathbf{o}}_{t-1})^{1-\mathbf{b}_t}, \tag{3}$$

where the indicator function $\mathbf{1}_{\mathbf{o}_t=o_{t-1}}$ used in Eq. 1 is now redefined as a degenerate probability distribution (Puterman, 1994):

$$P(\bar{\mathbf{o}}_t|\bar{\mathbf{o}}_{t-1}) = \begin{cases} 1 & \text{if } \bar{\mathbf{o}}_t = \bar{\mathbf{o}}_{t-1}, \\ 0 & \text{if } \bar{\mathbf{o}}_t \neq \bar{\mathbf{o}}_{t-1}. \end{cases}$$

and the joint distribution can be written as:

$$P(\bar{\tau}) = P(\mathbf{s}_0)P(\bar{\mathbf{o}}_0)P(\mathbf{a}_0|\mathbf{s}_0, \bar{\mathbf{o}}_0) \prod_{t=1}^{\infty} P(\mathbf{s}_t|\mathbf{s}_{t-1}, \mathbf{a}_{t-1})P(\mathbf{a}_t|\mathbf{s}_t, \bar{\mathbf{o}}_t)$$

$$\sum_{\mathbf{b}_t} P(\mathbf{b}_t|\mathbf{s}_t, \bar{\mathbf{o}}_{t-1})P(\bar{\mathbf{o}}_t|\mathbf{b}_t, \mathbf{s}_t, \bar{\mathbf{o}}_{t-1}) \tag{4}$$

Although the mixture master policy in Eq. 4 is MDP-formulated, as a mixture component within it, the master policy $P(\bar{\mathbf{o}}_t|\mathbf{s}_t)$ is still SMDP-formulated and hence cannot be updated by MDP-based algorithms. By marginalizing over the termination variable $\mathbf{b}_t$ in Eq. 4: $\sum_{\mathbf{b}_t} P(\mathbf{b}_t|\mathbf{s}_t, \bar{\mathbf{o}}_{t-1})P(\bar{\mathbf{o}}_t|\mathbf{b}_t, \mathbf{s}_t, \bar{\mathbf{o}}_{t-1})$, we propose the *Markovian master policy* $P(\bar{\mathbf{o}}_t|\mathbf{s}_t, \bar{\mathbf{o}}_{t-1})$ to model this marginal distribution explicitly:

$$P(\bar{\tau}) = P(\mathbf{s}_0)P(\bar{\mathbf{o}}_0)P(\mathbf{a}_0|\mathbf{s}_0,\bar{\mathbf{o}}_0)\prod_{t=1}^{\infty}P(\mathbf{s}_t|\mathbf{s}_{t-1},\mathbf{a}_{t-1})$$
$$P(\mathbf{a}_t|\mathbf{s}_t,\bar{\mathbf{o}}_t)P(\bar{\mathbf{o}}_t|\mathbf{s}_t,\bar{\mathbf{o}}_{t-1}) \tag{5}$$

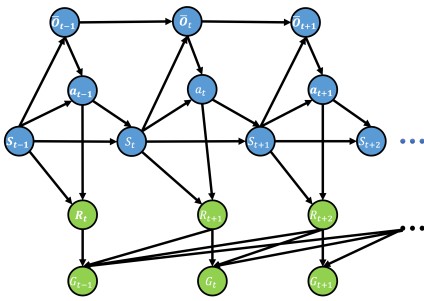

where Eq. 5 denotes the joint distribution of the PGM. In this formulation, $P(\bar{\tau})$ is actually an HMM with $\mathbf{s}_t, \mathbf{a}_t$ as observable random variables and $\bar{\mathbf{o}}_t$ as latent variables. We use $\pi^O(\mathbf{s}_t, \bar{\mathbf{o}}_{t-1}) = P(\bar{\mathbf{o}}_t|\mathbf{s}_t,\bar{\mathbf{o}}_{t-1})$ to denote the master policy and $\pi^A(\mathbf{s}_t, \mathbf{o}_t) = P(\mathbf{a}_t|\mathbf{s}_t,\bar{\mathbf{o}}_t)$ to denote the action policy. Figure 1 shows the PGM (Eq. 5).

**Figure 1:** Probabilistic Graphical Model (PGM) of *HiT-MDP*.

### 3.2 THE HIDDEN TEMPORAL MDPS (HIT-MDPS)

Given the PGM (Eq. 5, Figure 1), the Hidden Temporal MDPs (HiT-MDPs) family can be described by a tuple $M = \{\bar{\mathbb{S}}, \bar{\mathbb{A}}, r, P, \phi, \gamma\}$ where $\bar{\mathbb{S}} \doteq \mathbb{S} \times \bar{\mathbb{O}}$ is an augmented state space, $\bar{\mathbb{A}} \doteq \mathbb{A} \times \bar{\mathbb{O}}$ is an augmented action space, and $\phi = P(\bar{\mathbf{o}}_t|\bar{\mathbf{s}}_t) = P(\bar{\mathbf{o}}_t|\mathbf{s}_t,\bar{\mathbf{o}}_{t-1})$ is the emit function for hidden variables. The joint distribution of HiT-MDPs is factorized as Eq. 5 (derivations see Appendix C).

We define a partition function $\bar{B}(\bar{\mathbf{s}}, \bar{\mathbf{a}}) = \bar{B}(\{\bar{\mathbf{o}}_{t-1}, \mathbf{a}_t, \bar{\mathbf{o}}_t, \mathbf{s}_t\}) = (\mathbf{o}_{t-1}, \mathbf{a}_t, \mathbf{o}_t, \mathbf{s}_t)$ (proofs in Appendix C), which maps $\bar{\tau}$ to $\tau$, where $\bar{\tau} = \{\mathbf{s}_0, \bar{\mathbf{o}}_0, \mathbf{a}_0, \mathbf{s}_1, \bar{\mathbf{o}}_1, \mathbf{a}_1, \dots\}$ is the trajectory of the *HiT-MDP*. Therefore, by following the *Partition Function $\bar{B}$*, the dynamics of the *SMDP-Option* in Eq. 1 under the Surjection $\bar{B}$ is equivalent to *HiT-MDP* $P(\tau/\bar{B}) = P(\bar{\tau}/\bar{B})$. With $P(\tau/\bar{B}) = P(\bar{\tau}/\bar{B})$ in hand, to prove the equivalence between the *SMDP-Option* and *HiT-MDP*, we move on to prove both of them share the same expected reward. This is non-trivial since compared to the *SMDP-Option*, the MDP formulation introduces extra dependencies on $\bar{\mathbf{o}}$. However, in Appendix C, by exploiting conditional independencies we prove that they share the same expected return under the Surjection $\bar{B}$. Therefore, the SMDP-based option framework has an MDP-based equivalence:

**Theorem 3.1.** *By the definition of Bisimulation Relation, the SMDP-based option framework, which employs Markovian options, has an underlying MDP equivalence because:*

    1.                     $P(\tau/\bar{B}) = P(\bar{\tau}/\bar{B})$ and $\bar{B}$ is a Surjection.

    2.                     $r(\tau/\bar{B}) = r(\bar{\tau}/\bar{B})$

*Proof.* See Appendix C. □

Since the *master policy* $P(\bar{\mathbf{o}}_t|\mathbf{s}_t,\bar{\mathbf{o}}_{t-1})$ introduces one extra dependency on $\bar{\mathbf{o}}_{t-1}$, conventional Bellman equation which is derived by following the conventional value function $V[\mathbf{s}_t]$ no longer applies to *HiT-MDP* (see Appendix D). In order to derive the Bellman equation of *HiT-MDP*, we propose the novel *Markovian option-value function*, value functions with Markov dependencies (such as $\bar{\mathbf{o}}_{t-1}$). Specifically, rather than use the conventional value function $V[\mathbf{s}_t]$, let the *Markovian option-value function* $\bar{V}[\mathbf{s}_t, \bar{\mathbf{o}}_{t-1}]$ and the *option-action value function* $Q_A[\mathbf{s}_t, \bar{\mathbf{o}}_t, \mathbf{a}_t]$ be defined by:

$$\bar{V}[\mathbf{s}_t, \bar{\mathbf{o}}_{t-1}] = \mathbb{E}[G_t|\mathbf{s}_t, \bar{\mathbf{o}}_{t-1}] = \sum_{\bar{\mathbf{o}}_t} P(\bar{\mathbf{o}}_t|\mathbf{s}_t, \bar{\mathbf{o}}_{t-1})\mathbb{E}_{\mathbf{a}_t \sim \pi^A}[Q_A[\mathbf{s}_t, \bar{\mathbf{o}}_t, \mathbf{a}_t]], \tag{6}$$

$$Q_A[\mathbf{s}_t, \bar{\mathbf{o}}_t, \mathbf{a}_t] = \mathbb{E}[G_t|\mathbf{s}_t, \bar{\mathbf{o}}_t, \mathbf{a}_t] = r(s,a) + \mathbb{E}_{\mathbf{s}_{t+1} \sim \pi^A}[\sum_{l=1}^{} \gamma^l r_{t+l}] \tag{7}$$

As with the standard Q-function and value function, we can relate the Q-function to the *Markovian option-value function* at a future state via a *Hidden Temporal Bellman Operator* $\mathcal{T}^{\mathcal{H}}$.

**Theorem 3.2.** *The option-action value function Eq. 7 satisfies the Bellman Operator $\mathcal{T}^{\mathcal{H}}$*

$$\mathcal{T}^{\mathcal{H}}Q_A[\mathbf{s}_t, \bar{\mathbf{o}}_t, \mathbf{a}_t] = \mathbb{E}[G_t|\mathbf{s}_t, \bar{\mathbf{o}}_t, \mathbf{a}_t]$$
$$= r(s,a) + \gamma \sum_{\mathbf{s}_{t+1}} P(\mathbf{s}_{t+1}|\mathbf{s}_t, \mathbf{a}_t)\bar{V}[\mathbf{s}_{t+1}, \bar{\mathbf{o}}_t], \tag{8}$$

*where the Markovian option-value function given by Eq. 6. Proof.* See Appendix D.

We can obtain the *option-action value function* for any policy by repeatedly applying $\mathcal{T}^{\mathcal{H}}$ and the sequence converges to the optimal value function:

**Theorem 3.3.** *(Markovian Option Policy Evaluation Theorem). Assume that throughout our computation the $Q_A[\cdot, \cdot]$ and $\bar{V}[\cdot]$ are bounded and $\mathbb{A} < \infty$, the sequence $Q_A^k$ defined by $Q_A^{k+1} = \mathcal{T}^{\mathcal{H}} Q_A^k$ will converge to the option-action value function $Q_A^{\pi_A}$ as $k \to \infty$.*

*Proof.* As with the standard convergence results for policy evaluation (Sutton & Barto, 2018), by the definition of $\mathcal{T}^{\mathcal{H}}$ (Eq. 8) the *option-action value function* $Q_A^{\pi_A}$ is a fixed point. In Appendix D we show that $\mathcal{T}^{\mathcal{H}}$ is a contraction, and then Theorem 3.3 follows immediately. □

In Appendix D, we further prove that $\bar{V}[\mathbf{s}_t, \bar{\mathbf{o}}_{t-1}]$ is an unbiased estimation of $V[\mathbf{s}_t]$ and has a variance-reduction effect compared to the conventional value function. This property is empirically witnessed and further discussed in experiments (Section 5.1).

**Proposition 3.4.** *$\bar{V}[\mathbf{s}_t, \bar{\mathbf{o}}_{t-1}]$ is an unbiased estimation of $V[\mathbf{s}_t]$.*

**Proposition 3.5.** *The variance of $\bar{V}[\mathbf{s}_t, \bar{\mathbf{o}}_{t-1}]$ is up-bounded by $V[\mathbf{s}_t]$.*

*Proof.* See both Proposition 3.4 and 3.5's proof in Appendix D □

The theoretical analysis above presents a counterintuitive fact: the SMDP formulated option framework has an MDP formulated homomorphic equivalence (HiT-MDPs), and they both converge to the same optimal value function. One natural question to ask is that how could an MDP temporally extend executions of options? Note that the *Markovian master policy* $P(\bar{\mathbf{o}}_t|\mathbf{s}_t, \bar{\mathbf{o}}_{t-1})$ is a result of the marginalization over the termination variable $\mathbf{b}_t$ (See Eq. 5 and Eq. 1) and has an extra dependency on $\bar{\mathbf{o}}_{t-1}$). Therefore, the *Markovian master policy* acts like a distance measure. The decision of selecting an option $\bar{o}_t$ can be made by simply comparing which $\bar{o} \in \bar{\mathbf{o}}$ is closest to the vector $[\mathbf{s}_t, \bar{o}_{t-1}]$. Because a vector is closest to itself, this mechanism has a natural tendency to assign $\mathbf{o}_t$ to $o_{t-1}$, and thus extends $\bar{o}_{t-1}$'s execution. On the other hand, a significantly different state $\mathbf{s}_t$ will pull the distance far enough from $o_{t-1}$ and result in other options being assigned. The effectiveness of the *Markovian master policy* is also empirically addressed in Section 5.4.

## 4 LEARNING HiT-MDPs UNDER THE MAXIMUM ENTROPY FRAMEWORK

In this section, we propose a stable and sample efficient Maximum entropy Option Policy Gradient (MOPG) algorithm to learn HiT-MDPs. Learning temporal abstraction like options has been a long standing challenge (Sutton et al., 1999; Givan et al., 2003; Kolobov et al., 2012; Bacon et al., 2017). In theory (Sutton et al., 1999), options are not generally necessary for learning optimal policies since MDPs are sufficient enough. Therefore, standard option learning algorithms optimizing Eq. 1 often leads to sub-optimal results: either degenerate options (Harb et al., 2018) (short execution time) switching back-and-forth frequently, or dominant options (Zhang & Whiteson, 2019) (long execution time) executing through the whole episode. Although there are various attempts

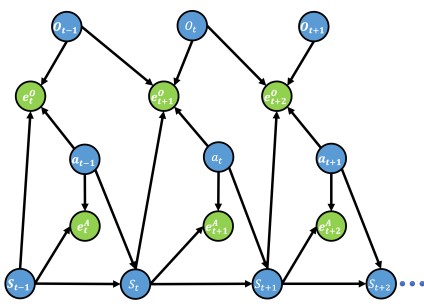

**Figure 2:** PGM of Eq. 5.

(Harb et al., 2018; Hyun et al., 2019; Smith et al., 2018) tackling this issue, yet they often introduce bias which accumulates along the length of the trajectory and result in sub-optimal solutions.

Recently maximum entropy reinforcement learning framework (Todorov, 2006; Ziebart et al., 2010; Haarnoja et al., 2017; 2018) has been witnessed success in encouraging exploration and improving sample complexity. With the PGM of the HiT-MDP in hand, we tackle the above issues by deriving the Maximum entropy Option Policy Gradient (MOPG), a stable algorithm for learning diversified options and preventing degeneracy. Following the control as inference framework (Levine, 2018; Haarnoja et al., 2018), we introduce the concept of "Optimality" (Todorov, 2006) into the HiT-MDP and formulate the conventional RL problems Eq. 5 as a probabilistic inference problem (Kappen et al., 2012). Specifically, we follow Levine (2018); Koller & Friedman (2009) and factorize Eq. 5 into another PGM (as shown in Figure 2):

$$q(\bar{\tau}, \mathbf{e}_{1:T}^A, \mathbf{e}_{1:T}^O) = P(\mathbf{s}_0)q(\bar{\mathbf{o}}_0)\prod_{t=1}^T q(\mathbf{e}_t^A|\mathbf{s}_t, \mathbf{a}_t)q(\mathbf{e}_t^O|\mathbf{s}_t, \mathbf{a}_t, \bar{\mathbf{o}}_t, \bar{\mathbf{o}}_{t-1})P(\mathbf{s}_{t+1}|\mathbf{s}_t, \mathbf{a}_t)q(\bar{\mathbf{o}}_t)q(\mathbf{a}_t)$$

$$\propto \underbrace{P(\mathbf{s}_0)\prod_{t=1}^T P(\mathbf{s}_{t+1}|\mathbf{s}_t, \mathbf{a}_t)}_{\text{Original Dynamics}} \underbrace{\prod_{t=1}^T q(\mathbf{e}_t^A|\mathbf{s}_t, \mathbf{a}_t)q(\mathbf{e}_t^O|\mathbf{s}_t, \mathbf{a}_t, \bar{\mathbf{o}}_t, \bar{\mathbf{o}}_{t-1})}_{\text{Variational Terms}}, \tag{9}$$

where $\mathbf{e} \in \{0, 1\}$ are observable binary "optimal random variables" (Levine, 2018). The agent is *optimal* at time step $t$ when $q(\mathbf{e}_t^A = 1|\mathbf{s}_t, \mathbf{a}_t)$ and $q(\mathbf{e}_t^O = 1|\mathbf{s}_t, \mathbf{a}_t, \bar{\mathbf{o}}_t, \bar{\mathbf{o}}_{t-1})$ (to keep notations uncluttered we use $e_t$ to denote $\mathbf{e}_t = 1$). To simplify the derivation, priors $q(\bar{\mathbf{o}})$ and $q(\mathbf{a})$ can be assumed to be uniform distributions without loss of generality (Levine, 2018). Note that Eq. 9 shares the same state-action dynamics with Eq. 5. With the optimal random variables $\mathbf{e}^O$ and $\mathbf{e}^A$, the conditional probability of a state-action $\{\mathbf{s}_t, \mathbf{a}_t\}$ pair that is optimal is defined as:

$$q(e_t^A|\mathbf{s}_t, \mathbf{a}_t) = \exp(r(\mathbf{s}_t, \mathbf{a}_t))[\int \exp(r(\mathbf{s}_t, \mathbf{a}_t))d\mathbf{a}]^{-1}, \tag{10}$$

which is an energy function that follows the Boltzmann distribution (Levine, 2018). This specific design facilitates recovering the value function at the latter structural variational inference stage. Based on the same motivation, the conditional probability of an option-state-action $\{\mathbf{o}_t, \mathbf{s}_t, \mathbf{a}_t, \mathbf{o}_{t-1}\}$ pair that is optimal is defined as,

$$q(e_t^O|\mathbf{s}_t, \mathbf{a}_t, \bar{\mathbf{o}}_t, \bar{\mathbf{o}}_{t-1}) = \exp(I[\bar{\mathbf{o}}_t|\mathbf{s}_t, \mathbf{a}_t, \bar{\mathbf{o}}_{t-1}])[\int \exp(I[\bar{\mathbf{o}}_t|\mathbf{s}_t, \mathbf{a}_t, \bar{\mathbf{o}}_{t-1}])d\mathbf{o}]^{-1}, \tag{11}$$

where the mutual-information $I[\bar{\mathbf{o}}_t|\mathbf{s}_t, \mathbf{a}_t, \bar{\mathbf{o}}_{t-1}]$ is chosen to be the energy function follows the Boltzmann distribution. This design choice arises from a fact that when the uniform prior assumption of $q(\mathbf{o})$ is relaxed the optimization introduces a mutual-information as a regularizer in the Evidence Lower BOund (ELBO) (Proof in Appendix E). When optimal random variables are observed, substituting Eq. 10 & 11 into Eq. 9, the conditional probability of any feasible trajectory $\bar{\tau}$ is:

$$q(\bar{\tau}|e_{1:T}) \propto \left[P(\mathbf{s}_0)\prod_{t=1}^{T}P(\mathbf{s}_{t+1}|\mathbf{s}_t, \mathbf{a}_t)\right]\exp(\sum_{t=1}^{T}r(\mathbf{s}_t, \mathbf{a}_t))\exp(\sum_{t=1}^{T}I[\bar{\mathbf{o}}_t|\mathbf{s}_t, \mathbf{a}_t, \bar{\mathbf{o}}_{t-1}]), \tag{12}$$

where optimal random variables $e_{1:T}$ are treated as observed random variables (evidences) and the trajectory $\bar{\tau}$ is treated as latent variables. Therefore, the problem of fitting the optimal trajectory is equivalent to maximizing ELBO: $\log q(e_{1:T}) \geq -D_{KL}[P(\bar{\tau})||q(\bar{\tau}|e_{1:T})]$ (Proof in Appendix E). This observation immediately gives rise to a structural variational inference solution as keeping the system dynamix $P(\mathbf{s}_0)\prod_{t=1}^{T}P(\mathbf{s}_{t+1}|\mathbf{s}_t, \mathbf{a}_t)$ in both Eq. 12 and Eq. 5 fixed while optimizing the action and master policies in the variational distribution of Eq. 5:

**Theorem 4.1.** *(Markovian Option Policy Improvement Theorem). The problem of learning optimal action and master policies can be simplified as shrinking the KL-Divergence:* $D_{KL}[P(\bar{\tau})||q(\bar{\tau}|e_{1:T})]$

$$\begin{aligned}\pi^{O*}, \pi^{A*} &= \arg\max_{\pi^O, \pi^A} -D_{KL}[P(\bar{\tau})||q(\bar{\tau}|e_{1:T})] \\ &= \arg\max_{\pi^O, \pi^A} \sum_t E_{\pi^O, \pi^A}[r(\mathbf{s}_t, \mathbf{a}_t) + I(\mathbf{o}_t|\mathbf{s}_t, \mathbf{a}_t, \mathbf{o}_{t-1}) + \mathcal{H}(\pi^O) + \mathcal{H}(\pi^A)],\end{aligned} \tag{13}$$

*where $\mathcal{H}(\cdot)$ denotes the entropy term. Proof.* See Appendix E.

With policy evaluation and improvement theorems, we further prove the convergence of MOPG by proposing the Markovian Option Policy Iteration Theorem (Appendix E). Standard option frameworks (as shown in Eq. 1 and Eq. 5) only consider maximizing the value function and have no constraints on qualities of options, e.g., what good options should behave (Harb et al., 2018). As a result, previous researches often lead to suboptimal results. In Eq. 44, the mutual-information is introduced as an intrinsic reward to encourage consecutive executions of options and thus prevent degenerate options, while entropy terms encourage explorations of options thus prevent dominant options.

Following value functions from Section 3.2, we directly optimize ELBO with respect to the variational distribution and derive the Maximum entropy Options Policy Gradients (MOPG) theorems:

**Theorem 4.2.** **Master Policy Gradient Theorem:** *Given a stochastic* master policy *which is differentiable w.r.t. its parameter vector $\theta_{\bar{o}}$, the gradient of the expected discounted return w.r.t. $\theta_{\bar{o}}$ is:*

$$\frac{\partial \bar{V}[\mathbf{s}_t, \bar{\mathbf{o}}_{t-1}]}{\partial \theta_{\bar{o}}} = \mathbb{E}[\frac{\partial P(\bar{\mathbf{o}}'|\mathbf{s}', \bar{\mathbf{o}})}{\partial \theta_{\bar{o}}}Q_O[\mathbf{s}', \bar{\mathbf{o}}'] + I(\bar{\mathbf{o}}'|\mathbf{s}, \mathbf{a}, \bar{\mathbf{o}}) + \mathcal{H}(\pi_{\theta_{\bar{o}}}^O) \mid \mathbf{s}_t, \bar{\mathbf{o}}_{t-1}], \tag{14}$$

*where $Q_O[\mathbf{s}_t, \bar{\mathbf{o}}_t] = \mathbb{E}[G_t|\mathbf{s}_t, \bar{\mathbf{o}}_t]$ (Appendix D Eq. 28). Proof.* See Appendix E

**Theorem 4.3.** **Action Policy Gradient Theorem:** *Given a stochastic action policy which is differentiable w.r.t. its parameter vector $\theta_a$, the gradient of the expected discounted return w.r.t. $\theta_a$ is:*

$$\frac{\partial Q_O[\mathbf{s}_t, \bar{\mathbf{o}}_t]}{\partial \theta_a} = \mathbb{E}[\frac{\partial P(\mathbf{a}|\mathbf{s}, \bar{\mathbf{o}})}{\partial \theta_a}Q_A[\mathbf{s}, \bar{\mathbf{o}}, \mathbf{a}] + \mathcal{H}(\pi_{\theta_a}^A) \mid \mathbf{s}_t, \bar{\mathbf{o}}_t]. \tag{15}$$

*Proof.* See Appendix E. To keep notations uncluttered, we use $\theta_{\bar{o}}$ to denote master policy's parameters $\pi_{\theta_{\bar{o}}}^O$ and $\theta_a$ to denote action policy's parameters $\pi_{\theta_a}^A$. The algorithm is summarized in Appendix F.

## 5 Experiments

In this section, we design experiments to answer five questions: (Q1) Whether MOPG can achieve better performance than other option variants and non-option baselines? (Q2) Does MOPG have a performance boost over other option variants in transfer learning settings? (Q3) Is HiT-MDP interpretable? (Q4) Whether MOPG can encourage consecutive executions of options while preventing dominant options problem?

For baselines, we follow DAC (Zhang & Whiteson, 2019)'s open source implementations and compare our algorithm with six baselines, five of which are option variants, *i.e.*, DAC+PPO, AHP+PPO (Levy & Shimkin, 2011), IOPG (Smith et al., 2018), PPOC (Klissarov et al., 2017), and OC (Bacon et al., 2017). The non-option baseline is PPO (Schulman et al., 2017). All baselines' parameters used by DAC remain unchanged other than the maximum number of training steps: MOPG only needs 1 million steps to converge rather than the 2 million used in DAC. For single task learning, experiments are conducted on all OpenAI Gym MuJoCo environments (10 environments) (Brockman et al., 2016a). For transfer learning, we follow DAC and run 6 pairs of transfer learning tasks based on DeepMind Control Suite (Tassa et al., 2020). Figures are plotted following DAC's protocol: curves are averaged over 10 independent runs and smoothed by a sliding window of size 20. Shaded regions indicate standard deviations. All experiments are run on an Intel® Core™ i9-9900X CPU @ 3.50GHz with a single thread and process. Our implementation details are summarized in Appendix G. For a fair comparison, we follow DAC and use four options in all implementations. Our code is available in supplemental materials.

### 5.1 Single-Task Learning (Q1)

To answer Q1, we compare MOPG against five different option variants (*i.e.*, DAC+PPO (Zhang & Whiteson, 2019), AHP+PPO (Levy & Shimkin, 2011), IOPG (Smith et al., 2018), PPOC (Klissarov et al., 2017) and OC (Bacon et al., 2017)) and PPO (Schulman et al., 2017). We observe that MOPG exhibits two different behaviors on infinite (Figure 3a) and finite (Figure 3b) horizon environments. Previous literatures (Klissarov et al., 2017; Smith et al., 2018; Harb et al., 2018; Zhang & Whiteson, 2019) find that option-based algorithms do not have advantages over hierarchy-free algorithms on single-task environments. Despite this, MOPG can still achieve comparable performance with state of the arts (Figure 3b). More importantly, on infinite horizon environments (Figure 3a), MOPG's performance significantly outperforms all baselines with respect to episodic return, convergence speed, variance between steps, and variance between 10 runs (Proposition 3.5). Since infinite and finite horizon environments are theoretically identical (Sutton & Barto, 2018), we do not have a theoretical explanation for this phenomenon. In Appendix A, we conceptually explain that this might be because conventional value functions are insufficient to approximate environments in which hidden variables **o** only affect rewards but not states. In conclusion, our experiment results show that MOPG is at least as effective as other option variants, but is significantly more sample efficient. Furthermore, it has significant advantages in infinite horizon environments.

### 5.2 Transfer Learning (Q2)

We run 6 pairs of transfer learning tasks based on DeepMind Control Suite (Tassa et al., 2020). Each pair contains two different tasks. Following the setting in DAC (Zhang & Whiteson, 2019), we train all models one million steps on the first task and then switch to the second task to further train another one million steps. Results are reported in Figure 4. On the transfer learning task, MOPG's

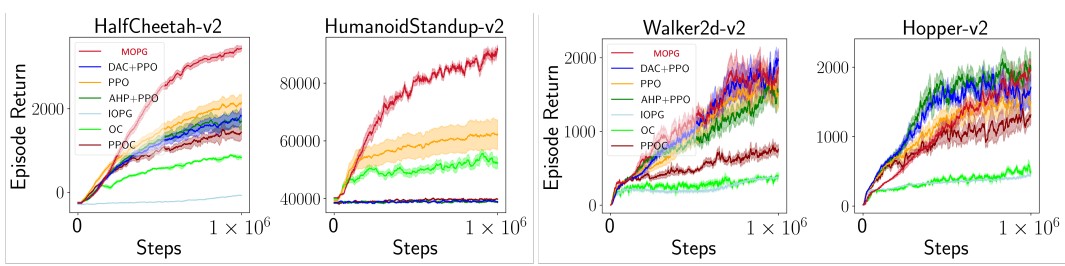

**(a)** Infinite horizon environments        **(b)** Finite horizon environments

**Figure 3:** Single-task episodic returns in 4 different environments (*i.e.*, HalfCheetah-v2, HumanoidStandup-v2, Walker2d-v2, and Hopper-v2). Results in all 10 environments are available in Appendix B.1.

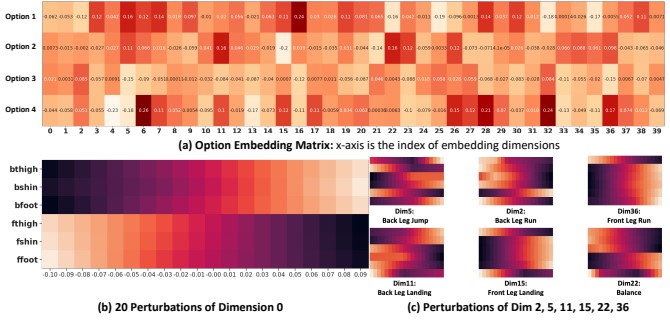

**Figure 4:** Transfer learning results on three pairs of tasks. All 6 pairs of results are available in Appendix. B.1

performance ranks the first in 5 out of 6 environments. This demonstrate MOPG's advantages in knowledge reuse and its performance is at least comparable with other option variants.

## 5.3 INTERPRETATION OF OPTION EMBEDDINGS (Q3)

Interpretability is a key property to apply RL agents in real-world applications. Our implementation follows the Skill-Action (SA) architecture (Li et al., 2020) and introduces transformer's decoder as a distance measure to learn option embeddings.

One unique advantage of HiT-MDP is that hidden variables $\bar{\mathbf{o}}$ are interpretable. In the implementation, the master policy is defined as $\pi^O = P(\bar{\mathbf{o}}_t | \mathbf{s}_t, \bar{\mathbf{o}}_{t-1}; \boldsymbol{W}_{\mathbf{o}})$ where $\boldsymbol{W}_{\mathbf{o}} = [\hat{\mathbf{o}}_1, ..., \hat{\mathbf{o}}_K]$ is an embedding matrix, $\hat{\mathbf{o}} \in \mathbb{R}^N$ is the embedding vector for an option with dimension $N$, and $K$ is the total number of options ($N = 40$ and $K = 4$ in figure 5(a)). More implementation details are described in Appendix G. Similar to word embeddings (Vaswani et al., 2017), option embeddings learn a semantic space of options with each dimension encodes a particular property and can be interpreted explicitly. As in the capsule network (Sabour et al., 2017), we first infer each embedding dimension's semantic by adding perturbations on to it $\hat{\mathbf{o}}_{[i,j]}^{\text{perturb}} = \epsilon_{[i]} + \hat{\mathbf{o}}_{[:,j]}$, where $\epsilon_i \in [-0.1, -0.09, ..., 0.09]$ is an array ranging from $-0.1$ to $0.09$ with an interval of $0.01$ and $j \in N$ is the $j$th dimension of the option embedding. We then sample primary actions $\mathbf{a}_{[:,j]} \sim \pi^A(\mathbf{s}_t, \mathbf{o}_{[i,j]}^{\text{perturb}})$ from perturbed options to observe how actions change along with perturbations.

For instance, Figure 5(b) visualizes how perturbations on dimension 0 of the option embedding affect the primary action. The Y-axis denotes actions defined by the HalfCheetah environment. The magnitude of Dim 0 have an opposite effect on the back leg and front leg: a larger value will increase the back leg's torque while decrease the front leg's, and vice versa. This means Dim 0 has a "focus point" property: it focuses torque on only one leg. Other dimensions in Figure 5(c) can be interpreted in the same manner. Once each dimension is understood, embeddings become straightforward to interpret by simply inspecting on which dimensions each embedding $\hat{\mathbf{o}}$ (Figure 5(a)) has significant weight. Due to space limitations, more details and GIFs are provided in Appendix B.3.

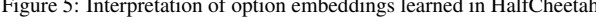

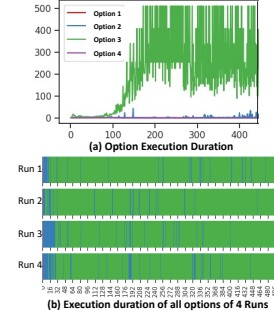

Figure 5: Interpretation of option embeddings learned in HalfCheetah

Figure 6: Options Composing Patterns

## 5.4 TEMPORAL EXTENSION (Q4)

One significant advantage of MOPG over other algorithms is that the intrinsic reward and entropy terms (Eq. 44) enable MOPG automatically learns to balance between degenerate and dominant options. In this subsection, we empirically demonstrate this in Figure 6 (experiment results from HalfCheetah, more details in Appendix B.2). At the start of training, all options' durations are short, while Option 3's duration quickly grows. This proves that MOPG can indeed temporally

extend an option. More than temporal extension, as shown in the video[1], MOPG learns to compose distinguishable options. Option 3 (green background in the video) is a running forward skill thus it is executed most of the time. Option 2 (blue background) is mainly used to recover from falling down thus its duration decreases with training. Therefore, the empirical study also shows that MOPG can compose disentangled options while preventing the dominant option problem.

## 6 RELATED WORKS

To discover options automatically, Sutton et al. (1999) proposed Intra-option Q-learning to update the master Q value function at every time step. However, all policies under this formulation are approximated implicitly using the Q-learning method. AHP (Levy & Shimkin, 2011) is proposed to unify the Semi-Markov process into an augmented Markov process and explicitly learn an "overall policy" by applying MDP-based policy gradient algorithms. However, their method for updating the master policy is still SMDP-style thus sample inefficient (Zhang & Whiteson, 2019). OC (Bacon et al., 2017) proposes a policy gradient based framework for explicitly learning intra-option policies and *termination function*s in an intra-option manner. However, for the master policy's policy gradients learning, OC remains SMDP-style. One closely related work is DAC (Zhang & Whiteson, 2019) which reformulates the option framework into two augmented MDPs. Under this formulation, all policies can be modeled explicitly and learned in MDP-style. However, DAC is not theoretically proven to be homomorphic equivalent to the SMDP-Option. Moreover, the reward function $r(\tau)$ used in Lemma 1 & 2 (Zhang & Whiteson, 2019) is undefined and makes the equivalence in proposition 1 (Zhang & Whiteson, 2019) does not hold. To the best of our knowledge, this is the first work proving MDPs' equivalence to option-induced SMDPs under the homomorphic equivalence framework.

Regarding network architectures, Attention Option Critic (AOC) (Chunduru & Precup, 2020) first introduces the attention mechanism into the option framework as a unique attention mask over state $\ddot{\mathbf{s}} = \ddot{W}_o \odot \mathbf{s}$ and enables learning interpretable options with unique activation field (e.g., initiation set) over the observation space. Unlike AOC, Section 5.3 follows the Skill-Action architecture (Li et al., 2020) which employs the attention mechanism over the joint space of the observation and option space. Regarding optimizing algorithms, Zhang & Whiteson (2019) pointed out that a large margin of the performance boost of DAC comes from the Proximal Policy Optimization (Schulman et al., 2017) (PPO). However, recent advances in probability inference based RL methods have also shown great success. Haarnoja et al. (2018) combined the idea of soft optimality (Haarnoja et al., 2017) from earlier control as inference frameworks (Todorov, 2006; Ziebart et al., 2010; Kappen et al., 2012) with the actor-critic architecture (Konda & Tsitsiklis, 2000) and proposed the state-of-the-art off-policy baseline SAC. Recent works show that the option framework trained under soft-optimality off-policy algorithms outperform on-policy methods (Wulfmeier et al., 2020). HiT-MDPs are general MDPs which can be trained by both on-policy and off-policy algorithms: the ELBO proposed in Section 4 can easily be extended to a SAC-like algorithm and remains to the future work.

## 7 CONCLUSIONS

In this paper, we propose the Hidden Temporal MDP (HiT-MDP), to the best of our knowledge, the first work proving option-induced MDP' is homomorphic equivalent to the standard SMDP Option framework. As for the learning algorithm, we propose the Maximum entropy Option Policy Gradients (MOPG) algorithm. Unlike conventional option learning algorithms that only maximize the value function without any constraint on the qualities of options, MOPG optimizes the Evidence Lower BOund (ELBO) with an intrinsic reward to prevent degenerate options and entropy terms to prevent dominant options. It is also theoretically proven fitting optimal trajectories without adding any bias.

On challenging *Mujoco* (Todorov et al., 2012; Brockman et al., 2016b; Tunyasuvunakool et al., 2020) environments, thorough empirical results demonstrate that under widely used configurations, HiT-MDP achieves comparable results to all baselines on finite horizon and transfer learning environments, and significantly outperforms all baselines on infinite horizon environments. We also demonstrate hidden temporal embeddings are interpretable, which is a key property for applying RL agents to real-world applications. Embeddings exist in various applied ML domains such as CV and NLP, our work shows that the idea also works for HRL and potentially lays the theoretical ground for building a large scale HRL foundation model.

---

[1]https://youtube.com/shorts/M06BPqit7l4?feature=share

ACKNOWLEDGEMENT

CL offers his sincerest gratitude to Shangtong Zhang, Pierre-luc Bacon and Doina Precup for their time and patience answering his questions. We deeply appreciate Shangtong Zhang and Jiayi Weng for their exceptional open source projects DeepRL https://github.com/ShangtongZhang/DeepRL (the original implementation in supplementary material and all experiment results in this paper are based on DeepRL) and tianshou https://github.com/thu-ml/tianshou (the latest github repo is based on tianshou).

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

## A  LEARNING OPTIONS AT MULTI-LEVELS OF GRANULARITY

Implementations of the option framework share some common limitations. When proposing the option framework, Sutton et al. (1999) expected that learning at multi-level of temporal abstraction should be in favor of faster convergence and better exploration. On the contrary, significant improvements on single task environments have not been witnessed in most option implementations (Klissarov et al., 2017; Smith et al., 2018; Harb et al., 2018; Zhang & Whiteson, 2019). To the best of our knowledge, MOPG is the first option implementation in which these properties are significantly witnessed but only on infinite horizon environments. In this section, we study this problem by explaining an intuition of why the value function is the main reason for this deficiency and how *deep wide value functions* could solve this problem.

The expectation of improvements of the option framework on single task environment builds on an assumption that, by exploiting hierarchical action and state space, an agent's searching space can be greatly reduced thus accelerates learning and improving exploration. However, as reported in section 5.4, most option frameworks suffer from "the dominant option problem" (Zhang & Whiteson, 2019) which prevents option frameworks from effectively learning hierarchy in action and state space as well as coordinating between options.

One intuition behind this problem is that conventional value functions $V[S_t]$ and $Q[S_t, O_t, A_t]$ make values depend on temporal latent variables indistinguishable (i.e. Although different options $o_1$ and $o_2$ results to different values, such as $V[S_t, O_{t-1} = o_1] = 10$ and $V[S_t, O_{t-1} = o_2] = -10$. Because they arrive at the same state $S_t$, they have identical values under conventional value function $V[S_t] = 0$). This deficiency makes option frameworks can only learn options at very coarse level thus fail to exploit hierarchical information. The solution might be using a *deep wide value function*: enabling the framework to learn fine-grained options at mutli-levels of granularity (deep) and making value functions depend on latent variables with longer (wide) dependencies (e.g. $V[S_t, O_{t-1}]$ and $Q[S_t, O_t, A_t, O_{t-1}]$).

To have a better understanding the importance of the deep wide value function, let us consider a simple environment which can be easily solved by $Q[s_t, a_t, a_{t-1}]$ but not $Q[s_t, a_t]$.

Suppose we are training a robot which only has a camera sensor to cook thanksgiving turkey. In this setting there are only two states: $\mathbb{S} = \{\text{Raw Turkey Image}, \text{Cooked Turkey Image}\}$. The robot's action space only consists of two actions: $\mathbb{A} = \{\text{Stuff turkey}, \text{Roast turkey}\}$. As for reward, if the robot roasted a stuffed turkey, then the reward is $10$. However, if the robot roasted an un-stuffed turkey, then the reward is $-10$. The stuff turkey action receives $0$ reward.

The difficulty in this environment is, since the robot only has a camera to capture an image of the turkey, it can only observes either {Raw Turkey Image} or {Cooked Turkey Image}. There is no way to look inside the turkey and see if the turkey is stuffed. Under this setting, a robot can never learn to first stuff a turkey and then roast it because $Q[\text{Raw Turkey Image}, \text{Stuff Turkey}] = Q[\text{Raw Turkey Image}, \text{Roast Turkey}] = 0$. Therefore, the robot can only randomly cook a turkey. However, this problem can be easily solved by using a deep wide value function $Q[S_t, A_t, A_{t-1}]$.

The core problem in this setting is, action has no effect on states, it only affects rewards. At the first glance this is a Partially Observed MDP (POMDP) problem since the state of whether the turkey is

stuffed is un-observed. This is true in all reinforcement learning settings without dependencies on latent variables. However, it goes much deeper in HRL settings.

In HRL, a common formulation is to estimate a latent variable $O$ to encode hierarchical information and makes the policy depends on it $P(A_t|S_t, O_t)$. Since $O$ is a latent variable, it is highly likely that at state $S_t$, different latent variable $P(A_t|S_t, O_t = o_x)$ and $P(A_t|S_t, O_t = o_y)$ emits the same action $A_t = A_1$, and thus makes the conventional value function indistinguishable between $o_x$ and $o_y$.

This phenomenon is especially common around the switching time step of two options: around switching point, states are usually compatible with both old and new options. Conventional value functions will be especially confused at those moments. This is exactly what we observed in Figure 5.3: overall, option 3 is executed consistently. However, there are some random switches to option 2. And the randomization is increased between around switching time steps. To explicitly show this, we visualized "Run4" into a video: `https://www.youtube.com/shorts/M06BPqit7l4?feature=share`. The option selection is very random at the beginning of the episode as well as around the switching point (the 16th second). These are exactly the most confusing moments of conventional value functions. Thanks to proposition 3.4 and 3.5, one unbiased solution might be employing the deep wide value function: the higher the order of the MDPs, the smaller the variance will be. We will explore this direction in our future work.

## B  EXPERIMENTS RESULTS

### B.1  PERFORMANCE

In this section we provide results for all ten OpenAI Gym Mujoco Environments (Brockman et al., 2016b; Todorov et al., 2012). Those environments can be classified into two categories: infinite horizon environments (i.e., HalfCheetah, Swimmer, HumanoidStandup and Reacher) and finite horizon environments (the other).

**Table 1:** Performance of Infinite Horizon Environments

|         | HalfCheetah | Swimmer | HumanoidStandup | Reacher |
|---------|-------------|---------|-----------------|---------|
| PPO     | 2143.6      | 59.9    | 62262.2         | -7.5    |
| DAC+PPO | 1830.1      | 85.0    | 38954.9         | -8.1    |
| AHP+PPO | 1701.7      | 86.7    | 38684.9         | -7.3    |
| PPOC    | 1441.2      | 43.6    | 39841.7         | -9.4    |
| OC      | 832.3       | 33.0    | 52352.7         | -15.3   |
| MOPG    | **3446.7**  | **107.8** | **91654.5**   | **-4.6** |

**Table 2:** Performance of Finite Horizon Environments

|         | Walker2d | Hopper | InvertedPendulum | InvertedDoublePendulum | Ant | Humanoid |
|---------|----------|--------|------------------|------------------------|-----|----------|
| PPO     | 1512.5   | 1489.9 | 939.9            | 7112.6                 | 1049.6 | 562.1 |
| DAC+PPO | **1968.0** | 1702.2 | **943.7**      | 5804.5                 | 985.8  | 487.6 |
| AHP+PPO | 1520.6   | **1993.6** | 940.0        | **7120.7**             | **1359.3** | **569.3** |
| PPOC    | 756.1    | 1308.1 | 936.2            | 7117.6                 | 429.4  | 483.9 |
| OC      | 391.9    | 487.6  | 207.1            | 2369.4                 | 433.4  | 475.1 |
| MOPG    | 1856.9   | 1955.3 | 906.5            | 6884.1                 | 907.4  | 528.7 |

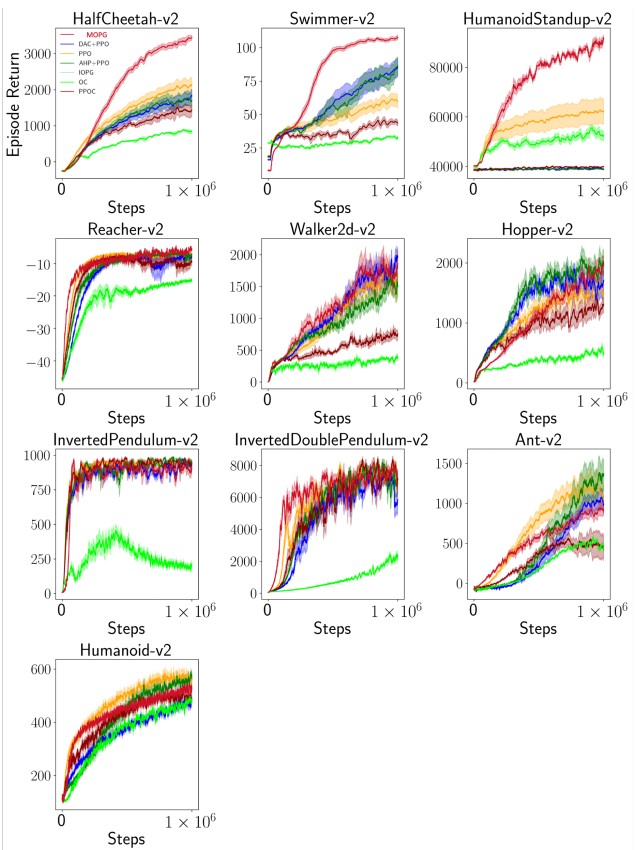

**Figure 5:** Performance of Ten OpenAI Gym MuJoCo Environments.

## B.2 TEMPORAL EXTENSION

In Figure 6, we plot the average duration of each option during 430 training episodes (each episode contains a trajectory of 512 time steps) of the HalfCheetah environment. In this environment, the agent learns to run half of a Cheetah by controlling 6 joints: back thigh, back shin, back foot, front thigh, front shin, and front foot. The faster the Cheetah runs forward, the higher return it gets from the environment. At the start of training, all options' durations are short. After the 100-th episode, Option 3's duration quickly grows. This answers the question Q4 in the main text that MOPG is able to temporally extending the execution of an option in the absence of the termination function.

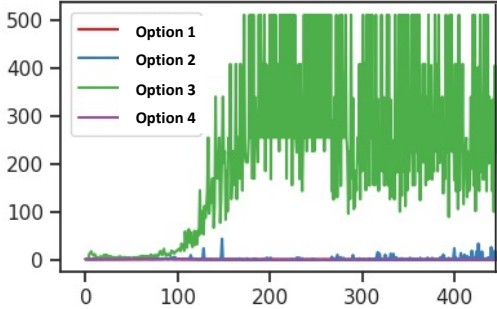

**Figure 6:** Duration of 4 options during 430 training episodes of HalfCheetah.

To illustrate how MOPG composes options, we take the HalfCheetah model trained after 1 million steps and independently run it 4 times (4 episodes. each episode contains 512 time steps). Option activation sequences of 4 runs are then plotted in Figure 7.

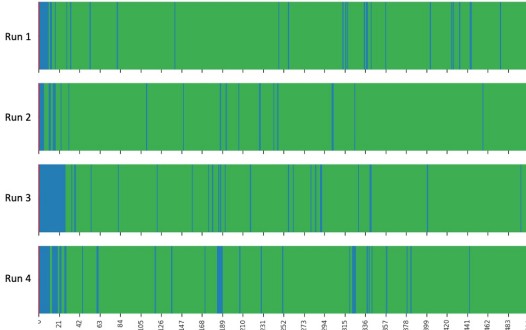

**Figure 7:** Activated option sequences of 4 independent HalfCheetah runs.

As we can see that there are some common patterns between all 4 independent runs. For example, all runs start with Option 1 and use Option 2 at the early stage. After executing Option 2 for a short period, they all switch to Option 3 which has longest durations in all 4 runs. From time to time they will fall back to Option 2 for short periods and quickly switch to Option 3 again. This pattern of coordination indicates that because of the intrinsic reward and entropy terms in Eq. 44, one significant advantages of the MOPG over other algorithms is that the ELBO objective function automatically learns to balance between degenerate options and dominant options. As shown in the video[2], Option 3 (green background in the video) is a running forward option thus it is executed most of the time. Option 2 (blue background) is mainly used to recover from falling down thus its duration decreases with training. Option 2 and Option 3 have completely different functionality. Therefore, empirical study also shows that MOPG is able to compose disentangled options while preventing dominant options problem.

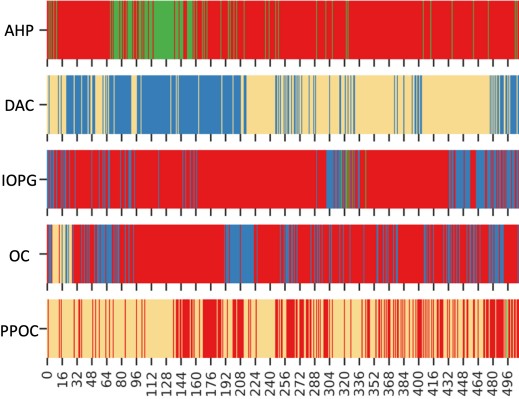

**Figure 8:** All option baselines' option durations for comparison in the HalfCheetah environment

One interesting question to ask is whether the performance boost of MOPG comes from increasing or decreasing of option frequencies compared to other option variants. In Figure 8 we plot one run of all option variants' option duration.

As shown in Figure 8, it appears to be that there is not a significant difference in terms of option duration between our method and other option variants. All of these methods are able to learn options which execute across various amount of time as well as composing those options. It is also worth to mention that comparing with Figure 3, there is not an apparent monotonically relation between option changing frequency and performance. Therefore, on simple environments like HalfCheetah, main performance difference comes with sample efficiency of training intra-option policies.

On the contrary, in harder environment like HumanoidStandupV2, there is a significant difference in terms of option composition between our method and other variants. As shown in Figure 3, all option

---

[2]https://youtube.com/shorts/M06BPqit7l4?feature=share

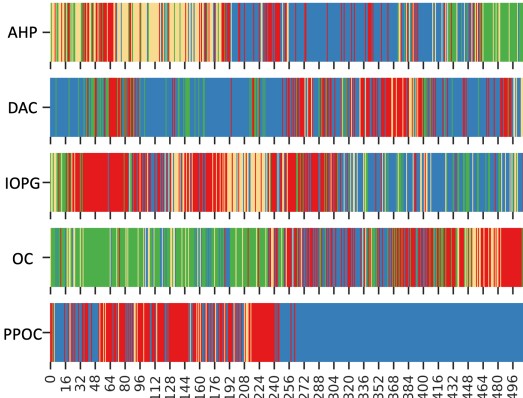

**Figure 9:** All option baselines failed to learn useful options in harder environment (HumanoidStandupV2)

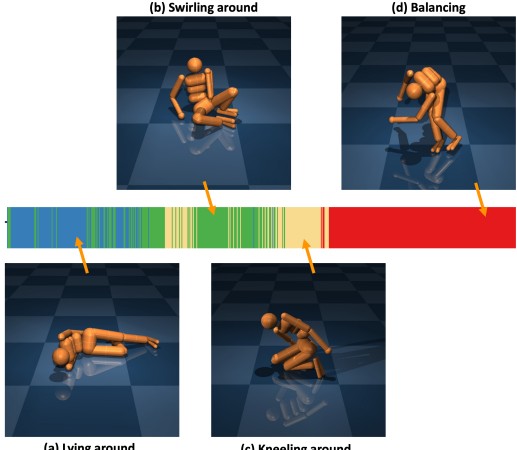

**Figure 10:** Illustration of our method's 4 Options composition in the HumanoidStandupV2 Environment.

variants except ours have a rather low score in HumanoidStandupV2. As shown in Figure 9, all of these option variants tends to quickly and randomly switching between options. This means that they failed to learn useful options which focus on specific sub-tasks with longer durations. This is because that standard option frameworks have tendencies to learn either degenerate (Harb et al., 2018) options (short execution time) that switching back-and-forth too frequently, or dominant options (Zhang & Whiteson, 2019) (long execution time) that executing through the whole episode. Both cases severely impair performance.

Figure 10 shows our method's options composition in HumanoidStandupV2. As shown in the figure that our method employs more options as the environment gets harder: it successfully learns 4 useful optons and has a clear options composition schedule between stages of standing up. For option (a), the humanoid is lying around and trying to sit up. For option (b) and option (c), these two options are more interchangeably used: option (b) is more sitting and swirling while option (c) is more trying to stand up and recover from failed trials. After the humanoid stands up, the agent constantly call option (d) which is running around and trying to balance itself. In conclusion, our method can learn distinguishable options and composition schedule in harder environment. Because of MOPG's maximum entropy framework, our method is able to adapt to environment's complexity in an end-to-end manner and is more robust to degenerate and dominant options.

One last thing worth to mention is that the number of options used is not only related to complexity of the environment, but also the coefficient of entropy terms. Figure 11 shows a histogram of the magnitude of entropy weights and how many options are used in total 20 runs. As we can see that as entropy weight increasing, more options are to be used since there is more randomness in option policy: when entropy weight is 0.01, all 20 runs only use 2 options; when entropy weight

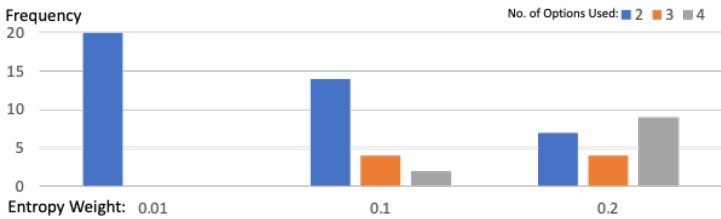

**Figure 11:** Histogram of entropy weights v.s. number of options used in HalfCheetah environment. The larger the entropy weight, more options will be used.

is 0.2, 9 of 20 runs use 4 options while only 7 runs use 2 options. Therefore, having an annealing training schedule of entropy weights might significantly boost performance of MOPG. However, to demonstrate the robustness of our method we fixed the entropy weight to 0.01 in all experiments.

### B.3 INTERPRETATION OF OPTION CONTEXT VECTORS

In this section we continue with the HalfCheetah model used in Section B.2 and demonstrate how to interpret option context vectors as well as option activation sequences (Figure 7). In HalfCheetah, the agent learns to run half of a Cheetah by controlling 6 joints: back thigh, back shin, back foot, front thigh, front shin, and front foot. The faster the Cheetah runs forward, the higher return it gets from the environment. We interpret option context vectors and activation patterns by first inspecting what property each dimension of the option context vector encodes (Figure 13). Once each dimension is understood, options (Figure 12) become straight forward to interpret by simply inspecting on which dimension (property) they have the most significant weights (Figure 14). These interpretations can further be taken to explain option activation patterns in Figure 7.

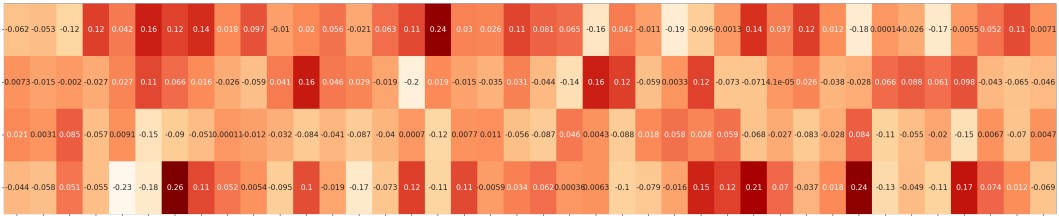

**Figure 12:** Heatmap of all 4 option context vectors

As the first step, we follow Sabour et al. (2017) to interpret what property each dimension of the option context vector in Figure 12 encodes by perturbing each dimension and decode perturbed option context vectors into primary actions. Specifically, we perturb one dimension by adding a range of perturbations $[-0.1, 0.09]$ by intervals of 0.01 onto it while keep the other dimensions fixed. After perturbation, each option context vector dimension has 20 perturbed vectors. We then use the action policy decoder to decode all those vectors into primary actions and see how the perturbation affects the primary action. As an illustration, we plot Dimension 4's all 20 perturbed results in Figure 13.

With visualization of perturbation results in hand, we can interpret what property each dimension encode by inspecting relationships between perturbations and primary actions. In Figure 13, as an example, it is clear that changes on Dim 4 has the same direction: as the magnitude of Dim 4 increase, all actuators move towards the same direction. This Dim can be seen as having an acceleration of running forward effect.

Once we know how to interpret one dimension, we can move on to interpret the whole option context vector. Since Option 1 and Option 2 are two main options employed in Figure 7, here we provide an example of how to interpret them. Figure 12 shows that Option 1 has significant values on dimension 11, 15 and 22. Option 2 is significant on dimension 2, 5 and 36. We demonstrate these dimensions in the same manner as Figure 13 below:

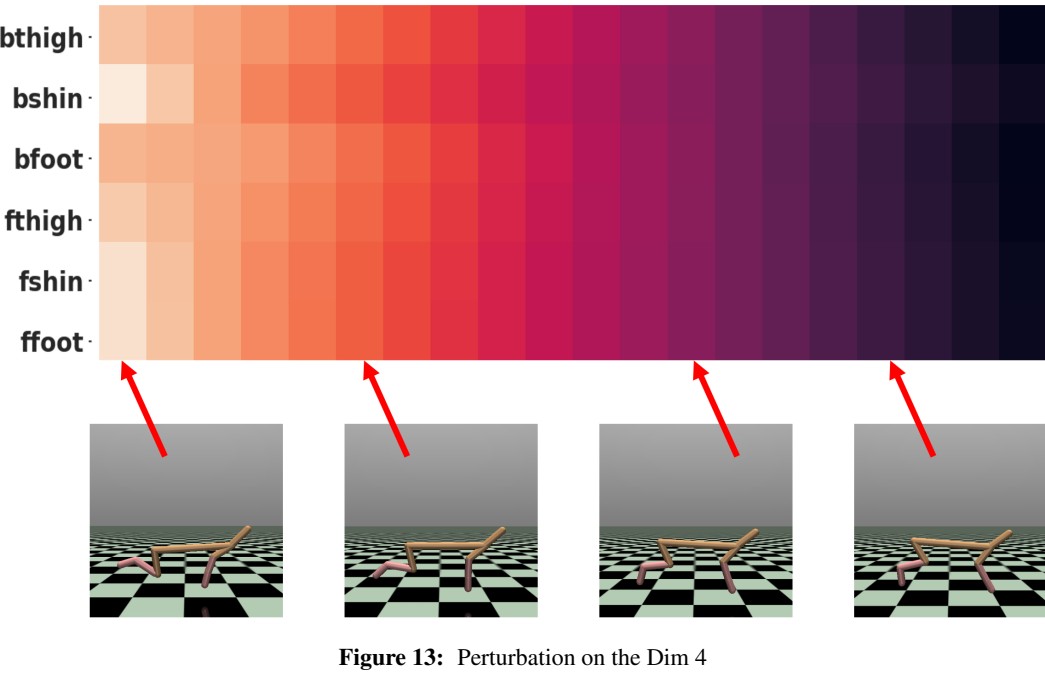

**Figure 13:** Perturbation on the Dim 4

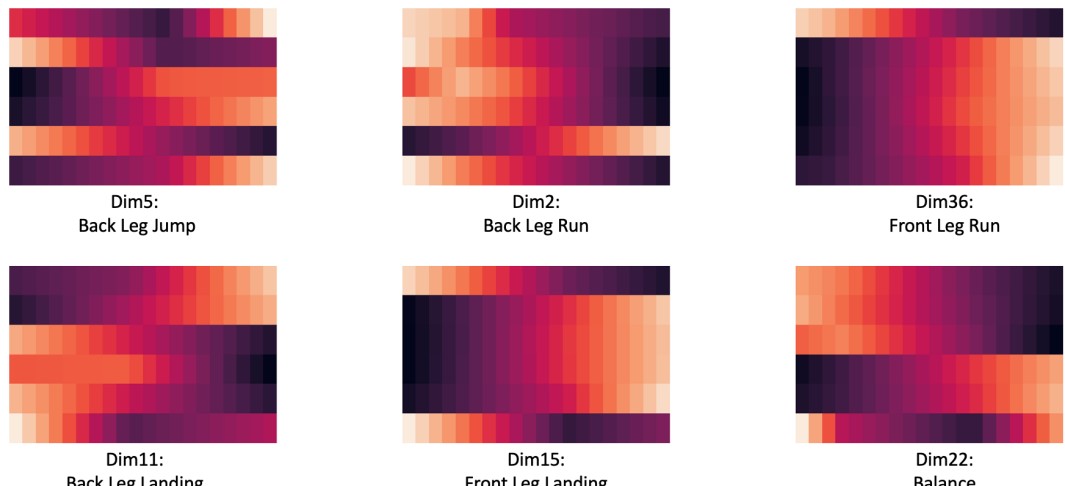

**Figure 14:** Interpretation of Option 1 and Option 2

Subfigures in Figure 14 can be interpreted in the same manner as Figure 13. As an example, from Figure 12 we can see that Option 1 has a significant small value on Dim 11. In Figure 14, it shows that a smaller Dim 11 will twist the front leg forward and back foot forward while twist back thigh, back shin backward. Composition of these movements is a back leg landing property. Similarly, we can interpret that Dim 15 is a front leg landing property and Dim 22 is a balancing property. Therefore, Option 1 is focusing on landing from all positions.

Unlike other option context vectors which have apparent focusing dimensions, Option 2 has a rather balanced option context vector. It has no apparently dominant dimension. It only has slightly more significant values on Dim 2, 5, 36, which are focusing on jumping and running properties. Therefore, Option 2 is more like an "all-weather" option: it is a option having very balanced properties with a slightly demonstration on running and jumping.

Interpretations of Option 1 and 2 above can then be taken to understand option activation patterns in Figure 7: as an all-weather option, Option 2 is the most frequently executed one and has the

longest duration. From time to time, when the Cheetah needs to land and balance itself, Option 1 will be executed. However, since landing option does not provide power of moving forward and thus has lower returns to continue, once the body is balanced the Cheetah will quickly stop Option 1's execution and keep running with Option 2.

### B.4 TRANSFER LEARNING RESULTS

Below are statistics for Deepmind Control Suite (Tunyasuvunakool et al., 2020) transfer learning environments.

**Table 3:** Performance of Deepmind Control Suite Transfer Learning Environments

|         | CartPole | Reacher | Cheetah | Fish  | Walker1 | Walker2 |
|---------|----------|---------|---------|-------|---------|---------|
| PPO     | 829.7    | 327.6   | 73.0    | 287.9 | 231.8   | 72.2    |
| DAC+PPO | 970.8    | 517.2   | 211.2   | 505.4 | **590.3** | 360.5 |
| AHP+PPO | 966.5    | 395.2   | 167.4   | 357.9 | 362.1   | 143.2   |
| PPOC    | 942.1    | 400.1   | 72.7    | 336.7 | 236.6   | 80.9    |
| OC      | 106.1    | 19.4    | 100.6   | 286.6 | 356.3   | 238.7   |
| MOPG    | **974.1** | **675.3** | **233.8** | **562.1** | 473.8 | **403.0** |

## C   AN MDP EQUIVALENCE TO THE SMDP OPTION FRAMEWORK

In this section, we show that the the conventional Semi-Markov Decision Problem (SMDP) option framework which employs Markovian options has an option-induced MDP equivalence. We first introduce the conventional SMDP formulated option framework (SMDP-Option). In Section C.2 we follow Bishop (2006)'s method and formulate the dynamics of the SMDP-Option framework as an Hidden Markov Model (HMM) style Probability Graphical Model (PGM) (Koller & Friedman, 2009). With the PGM and its conditional independence relationships (Chapter 8.2.1 (Bishop, 2006)) in hand, in Section C.3 we move on to propose the Hidden Temporal MDPs (HiT-MDP) and prove its equivalence to the SMDP-Option under the definition of homomorphic equivalence (Ravindran, 2003). To the best of our knowledge, this is the first work discovering the option framework's MDP equivalence.

Following Bishop (2006)'s notation, we use $A$, $B$ and $C$ to denote three non-overlapping sets of arbitrarily many random variables. Sets $A$ and $B$ are conditional independent on set $C$ if $P(A, B|C) = P(A|C)P(B|C)$, denoted as $A \perp\!\!\!\perp B \mid C$. We mainly use head-to-tail conditional independence properties (Chapter 8.2.1 (Bishop, 2006)) in this section.

### C.1   THE SMDP FORMULATED OPTION FRAMEWORK

Sutton et al. (1999) proposed the option framework to demonstrate the temporal abstraction problem. Following Bishop (2006)'s notation, we use bolded letter $\mathbf{s} \in \mathbb{S}$ to denote a random variable and normal letter s to denote its realization. Without special clarification, a random vector can have either a vector of continuous or discrete entries. A scalar o $\in \mathbb{Z}$ denotes the index of an option where $\mathbb{O} \subseteq \{1, 2, \ldots, M\}$ and $M$ is the number of options. An Markovian option is a triple $(\mathbb{I}_o, P_o(\mathbf{a}|\mathbf{s}), P_o(\mathbf{b}|\mathbf{s}))$ in which $\mathbb{I}_o \subseteq \mathbb{S}$ is an initiation set where the option $o$ can be initiated. $P_o(\mathbf{a}|\mathbf{s}) : \mathbb{S} \to \mathbb{A}$ is the intra-option policy which maps environment states $\mathbf{s} \in \mathbb{S}$ to an action vector $\mathbf{a} \in \mathbb{A}$. $P_o(\mathbf{b}|\mathbf{s}) : \mathbb{S} \to \mathbb{Z}_2$ is a termination function where b is a binary random variable. It is used to determine whether to terminate ($\mathbf{b} = 1$) the policy $P_o(\mathbf{a}|\mathbf{s})$ or not ($\mathbf{b} = 0$). Conventionally, $\beta_o = P_o(\mathbf{b} = 1|\mathbf{s})$. Since an option's execution may persist over a variable period of time, a set of options' execution together with its value functions constitutes a Semi-Markov Decision Problem (SMDP) (Puterman, 1994). When an old option is terminated, a new option will be sampled from the master policy (policy-over-options) $o \sim P(o_{t+1}|\mathbf{s}_{t+1}) : \mathbb{S} \to \mathbb{O}$.

A *master policy* $\pi(\mathbf{o}|\mathbf{s}) = P(\mathbf{o}|\mathbf{s})$ where $\mathbf{o} \in \mathbb{O}$ is used to sample which option will be executed. Note that we use the bold-case $\mathbf{o}$ to denote unrealized random variables and the light-italic-case $o$ to denote a realized instantiation. Conventionally, the execution of an option employs the call-and-return

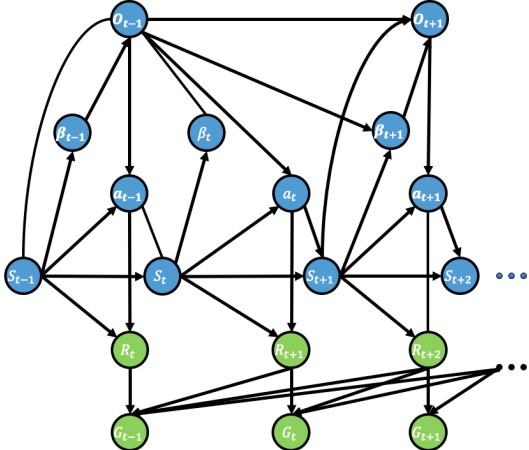

**Figure 15:** An Illustration of the SMDP Option Framework. An option $o_{t-1}$ is selected by master policy $P(o_{t-1}|s_{t-1})$ at time step $t-1$. At time step $t$, termination function $\beta_{o_{t-1}}(s_t)$ determines to continue option $o_{t-1}$. So that there is no random variable $o_t$ at time step $t$ compared to there are random variables $\mathbf{o}$ at every time step in MDP formulation (figure 16).

model (Sutton et al., 1999): at time step $t$, an agent either continues the previously executed option $\mathbf{o}_{t-1} = o$ with probability $P_o(\mathbf{b} = 0|\mathbf{s})$ and sets $\mathbf{o}_t = \mathbf{o}_{t-1} = o$, or terminates $o$ with probability $P_o(\mathbf{b} = 1|\mathbf{s})$ and samples a new option $\mathbf{o}_t$ from the master policy $P(\mathbf{o}_t|\mathbf{s}_t)$. Therefore, the dynamics (stochastic process) of the option framework is written as:

$$P(\tau) = P(\mathbf{s}_0)P(\mathbf{o}_0)P_{o_0}(\mathbf{a}_0|\mathbf{s}_0)\prod_{t=1}^{\infty} P(\mathbf{s}_t|\mathbf{s}_{t-1}, \mathbf{a}_{t-1})P_{o_t}(\mathbf{a}_t|\mathbf{s}_t)[(1-\beta_o)\mathbf{1}_{\mathbf{o}_t=o_{t-1}} + \beta_o P(\mathbf{o}_t|\mathbf{s}_t)].$$

$$= P(\mathbf{s}_0)P(\mathbf{o}_0)P_{o_0}(\mathbf{a}_0|\mathbf{s}_0)\prod_{t=1}^{\infty} P(\mathbf{s}_t|\mathbf{s}_{t-1}, \mathbf{a}_{t-1})P_{o_t}(\mathbf{a}_t|\mathbf{s}_t)$$

$$[P_{o_{t-1}}(\mathbf{b}_t = 0|\mathbf{s}_t)\mathbf{1}_{\mathbf{o}_t=o_{t-1}} + P_{o_{t-1}}(\mathbf{b}_t = 1|\mathbf{s}_t)P(\mathbf{o}_t|\mathbf{s}_t)]. \tag{16}$$

where $\tau = \{\mathbf{s}_0, \mathbf{o}_0, \mathbf{a}_0, \mathbf{s}_1, \mathbf{o}_1, \mathbf{a}_1, \ldots\}$ denotes the trajectory of the option framework. $\mathbf{1}$ is an indicator function and is only true when $\mathbf{o}_t = o_{t-1}$ (notice that $o_{t-1}$ is the realization at $\mathbf{o}_{t-1}$). For clarity reasons, we use $P_o(\mathbf{b} = 1|\mathbf{s})$ instead of $\beta_o$ which is widely used in previous option literatures (e.g. (Sutton et al., 1999; Bacon et al., 2017)). Therefore, under this formulation the option framework is defined as a Semi-Markov process since the dependency on an activated option $o$ can cross a variable amount of time (Sutton et al., 1999).

## C.2 An MDP formulation of the Option Framework

We follow Bishop (2006)'s formulation of mixture distribution and Probabilistic Graphical Models (PGMs). By introducing option variables as latent variables and adding extra dependencies into the termination function and master policy, we show that the conventional SMDP version of the option framework (Bacon et al., 2017; Sutton & Barto, 2018; Sutton et al., 1999; Harb et al., 2018; Zhang & Whiteson, 2019) can be re-formulated into an MDP formulation. We first follow Bishop (2006)'s formulation of mixture distributions and redefine the option random variable $\mathbf{o} \in \mathbb{O} = \{1, 2, \ldots, K\}$, which was originally defined as an integer index, but now as a $K$-dimensional one-hot vector $\bar{\mathbf{o}} \in \bar{\mathbb{O}} = \{0, 1\}^K$ where $K$ is the number of options, and each entry $\mathbf{o} \in \{0, 1\}$ is a binary random variable. $P(\bar{\mathbf{o}}_t|\mathbf{s}_t)$ denotes the probability distribution over one-hot vector $\bar{\mathbf{o}}$ at time step $t$ conditioned on state $\mathbf{s}_t$. $P(\bar{\mathbf{o}}_t = o_t|\mathbf{s}_t)$ denotes a probability entry (a scalar value) of the random variable $\bar{\mathbf{o}}_t$ with a realization at time step $t$ where $\mathbf{o}_t = 1$ and $\mathbf{o} \in \bar{\mathbf{o}}_t/\mathbf{o}_t = 0$.

In figure 16, $\mathbf{s} \in \mathbb{S}$, $\bar{\mathbf{o}} \in \bar{\mathbb{O}}^K$, $\mathbf{b} \in \mathbb{B}^K$ and $\mathbf{a} \in \mathbb{A}$, denote the state, option, termination and action random variable respectively. $\bar{\mathbf{o}}$ is an $K$-dimensional one-hot vector and $\mathbf{b}$ is an $K$-dimensional binary vector where each entry $\mathbf{b} \in \{0, 1\}$. $K$ is the number of options. $R_{t+1}$ is the actual reward received from the environment after executing action $\mathbf{a}_t$ in state $\mathbf{s}_t$. $G_t = R_{t+1} + \gamma R_{t+2} + \gamma^2 R_{t+3} \cdots$ is the discounted expected return where $\gamma \in \mathbb{R}$ is a discount factor.

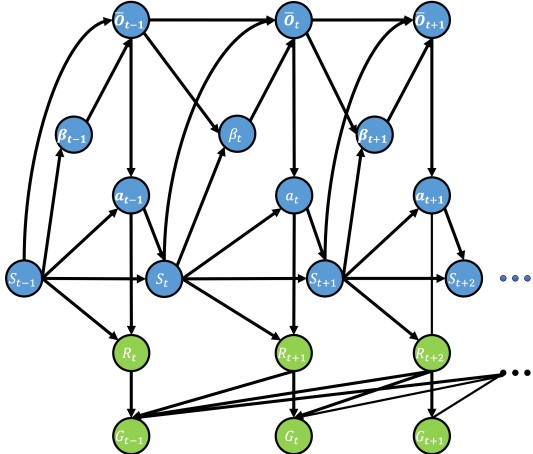

**Figure 16:** PGM of the MDP Option Framework. Notice the difference to Figure 15. Figure 15 is SMDP formulated, suppose the agent decides to continue the execution of $o_{t-1}$, then the random variable $\mathbf{o}_t$ does not exist, and thus $\mathbf{o}_{t+1}$ depends on the random variable $\mathbf{o}_{t-1}$ directly. However, here the PGM is MDP formulated, and the random variable $\bar{\mathbf{o}}$ exists at every time step.

The termination policy distribution $P(\mathbf{b}_t|\mathbf{s}_t, \bar{\mathbf{o}}_{t-1}) : \mathbb{S} \times \bar{\mathbb{O}} \to \mathbb{B}$ can be formulated as a mixture distribution[3] conditioned on option vector (the one-hot vector) $\bar{\mathbf{o}}_{t-1}$ and state $\mathbf{s}_t$.

$$P(\mathbf{b}_t|\mathbf{s}_t, \bar{\mathbf{o}}_{t-1}) = \prod_{i \in \bar{\mathbf{o}}_{t-1}} P_i(\mathbf{b}_t|\mathbf{s}_t)^i. \tag{17}$$

Because each option has its own **termination policy** $P_o(\mathbf{b}|\mathbf{s})$, with a slightly abuse of notation, in equation (17) we use $P(\mathbf{b}_t|\mathbf{s}_t, \bar{\mathbf{o}}_{t-1})$ to denote the termination policy activated at time step $t$ by previous chosen option $\bar{\mathbf{o}}_{t-1}$. To keep notation uncluttered, we use $\beta_t = P(\mathbf{b}_t = 1|\mathbf{s}_t, \bar{\mathbf{o}}_{t-1})$ to denote the probability of option $\bar{\mathbf{o}}_{t-1}$ terminates at time step $t$ and $(1 - \beta_t) = P(\mathbf{b}_t = 0|\mathbf{s}_t, \bar{\mathbf{o}}_{t-1})$ to denote the probability of continuation.

Conventionally, master policy (Zhang & Whiteson, 2019) (also called "policy-over-options" (Sutton et al., 1999; Bacon et al., 2017))) is defined as:

$$P(\bar{\mathbf{o}}_t|\mathbf{s}_t). \tag{18}$$

Similarly, we propose a novel **mixture master policy** as a mixture distribution[4]:

$$P(\bar{\mathbf{o}}_t|\mathbf{s}_t, \mathbf{b}_t, \bar{\mathbf{o}}_{t-1}) = P(\bar{\mathbf{o}}_t|\mathbf{s}_t)^{\mathbf{b}_t} P(\bar{\mathbf{o}}_t|\bar{\mathbf{o}}_{t-1})^{1-\mathbf{b}_t}, \tag{19}$$

where $P(\bar{\mathbf{o}}_t|\bar{\mathbf{o}}_{t-1})$ is a degenerated probability distribution (Puterman, 1994)

$$P(\bar{\mathbf{o}}_t|\bar{\mathbf{o}}_{t-1}) = \begin{cases} 1 & \text{if } \bar{\mathbf{o}}_t = \bar{\mathbf{o}}_{t-1}, \\ 0 & \text{if } \bar{\mathbf{o}}_t \neq \bar{\mathbf{o}}_{t-1}. \end{cases} \tag{20}$$

As shown in equation (19), the master policy only exists when $\mathbf{b}_t = 1$ the option terminates. Therefore, PPOC (Klissarov et al., 2017) uses inaccurate gradients for updating the master policy during an option's execution.

According to the conditional dependency relationships in PGM (figure 16), the joint probability distribution of $\bar{\mathbf{o}}_t$ and $\mathbf{b}_t$ can be written as:

$$P(\bar{\mathbf{o}}_t, \mathbf{b}_t|\mathbf{s}_t, \bar{\mathbf{o}}_{t-1}) = P(\mathbf{b}_t|\mathbf{s}_t, \bar{\mathbf{o}}_{t-1})P(\bar{\mathbf{o}}_t|\mathbf{s}_t, \mathbf{b}_t, \bar{\mathbf{o}}_{t-1}), \tag{21}$$

---

[3]Different from conventional formulation which only depends on state $\mathbf{s}_t$, our termination function has an extra dependence on $\bar{\mathbf{o}}_{t-1}$

[4]Different from conventional formulation which only depends on state $\mathbf{s}_t$, our mixture master policy has extra dependencies on $\bar{\mathbf{o}}_{t-1}$ and $\mathbf{b}_t$

and the marginal probability distribution can be written as:

$$P(\bar{\mathbf{o}}_t|\mathbf{s}_t, \bar{\mathbf{o}}_{t-1}) = \sum_{\mathbf{b}_t} P(\mathbf{b}_t|\mathbf{s}_t, \bar{\mathbf{o}}_{t-1})P(\bar{\mathbf{o}}_t|\mathbf{s}_t, \mathbf{b}_t, \bar{\mathbf{o}}_{t-1}) \tag{22}$$

$$= P(\mathbf{b}_t = 0|\mathbf{s}_t, \bar{\mathbf{o}}_{t-1})P(\bar{\mathbf{o}}_t|\bar{\mathbf{o}}_{t-1}) + P(\mathbf{b}_t = 1|\mathbf{s}_t, \bar{\mathbf{o}}_{t-1})P(\bar{\mathbf{o}}_t|\mathbf{s}_t)$$

$$= (1 - \beta_t)P(\bar{\mathbf{o}}_t|\bar{\mathbf{o}}_{t-1}) + \beta_t P(\bar{\mathbf{o}}_t|\mathbf{s}_t)$$

$$= (1 - \beta_t)\mathbf{1}_{\bar{\mathbf{o}}_t=\bar{\mathbf{o}}_{t-1}} + \beta_t P(\bar{\mathbf{o}}_t|\mathbf{s}_t).$$

The **intra-option (action) policy** distribution can also be formulated as a mixture distribution

$$P(\mathbf{a}_t|\mathbf{s}_t, \bar{\mathbf{o}}_t) = \prod_{i \in \bar{\mathbf{o}}_t} P_i(\mathbf{a}_t|\mathbf{s}_t)^i. \tag{23}$$

Therefore, the dynamics of the PGM in figure 16 can be written as:

$$P(\bar{\tau}) = P(\mathbf{s}_0)P(\bar{\mathbf{o}}_0)P(\mathbf{a}_0|\mathbf{s}_0, \bar{\mathbf{o}}_0)$$

$$\prod_{t=1}^{\infty} P(\mathbf{s}_t|\mathbf{s}_{t-1}, \mathbf{a}_{t-1})P(\mathbf{a}_t|\mathbf{s}_t, \bar{\mathbf{o}}_t) \sum_{\mathbf{b}_t} P(\mathbf{b}_t|\mathbf{s}_t, \bar{\mathbf{o}}_{t-1})P(\bar{\mathbf{o}}_t|\mathbf{b}_t, \mathbf{s}_t, \bar{\mathbf{o}}_{t-1}) \tag{24}$$

where $P(\bar{\tau}) = P(\mathbf{s}_0, \bar{\mathbf{o}}_0, \mathbf{a}_0, \mathbf{s}_1, \mathbf{b}_1, \bar{\mathbf{o}}_1, \mathbf{a}_1, \ldots)$ denotes the joint distribution of the PGM. Notice that under this formulation, $P(\tau)$ is actually an HMM with $\mathbf{s}_t, \mathbf{a}_t$ as observable random variables and $\mathbf{b}_t, \bar{\mathbf{o}}_t$ as latent variables.

It is worth to mention that equation (20) is essentially the indicator function $\mathbf{1}_{\bar{\mathbf{o}}_t=\bar{\mathbf{o}}_{t-1}}$ used in conventional SMDP option framework papers and the last line in equation (22) is identical to transitional probability distribution in their formulation. However, as we show in this section, by adding latent variables $\bar{\mathbf{o}}_{t-1}$ and introducing the dependency between $\bar{\mathbf{o}}_t$ and $\mathbf{b}_t$, our formulation is essentially an HMM. It opens the door to introduce many well developed PGM algorithms such as message passing (Forney, 1973) and variational inference (Hoffman et al., 2013) to the reinforcement learning framework. As we show below, the nice conditional independence relationships enjoyed by this model also enable us to prove the equivalence between the option framework's SMDP and MDP formulation.

## C.3   HIDDEN TEMPORAL MDPS (HIT-MDPS)

In this section we propose the Hidden Temporal MDPs (HiT-MDPs) and prove its equivalence under the definition of homomorphic equivalence. Following notations from Section C.2, the Hidden Temporal MDPs (HiT-MDPs) family can be described by a tuple $\bar{M} = \{\bar{\mathbb{S}}, \bar{\mathbb{A}}, r, P, \phi, \gamma\}$ where $\bar{\mathbb{S}} \doteq \mathbb{S} \times \bar{\mathbb{O}}$ is an augmented state space, $\bar{\mathbb{A}} \doteq \mathbb{A} \times \bar{\mathbb{O}}$ is an augmented action space, and $\phi = P(\bar{\mathbf{o}}_t|\bar{\mathbf{s}}_t) = P(\bar{\mathbf{o}}_t|\mathbf{s}_t, \bar{\mathbf{o}}_{t-1})$ is the emit function for hidden variables. The joint distribution of HiT-MDPs is:

$$P(\bar{\tau}) = P(\bar{\mathbf{s}}_1)\prod_{t=1}^{\infty} P(\bar{\mathbf{s}}_{t+1}, \bar{\mathbf{a}}_t|\bar{\mathbf{s}}_t) = P(\bar{\mathbf{o}}_0, \mathbf{s}_1)\prod_{t=1}^{\infty} P(\mathbf{s}_{t+1}, \mathbf{a}_t, \bar{\mathbf{o}}_t|\mathbf{s}_t, \bar{\mathbf{o}}_{t-1}), \tag{25}$$

$$= P(\bar{\mathbf{o}}_0)P(\mathbf{s}_1)\prod_{t=1}^{\infty} P(\mathbf{s}_{t+1}|\mathbf{s}_t, \mathbf{a}_t)P(\mathbf{a}_t|\mathbf{s}_t, \bar{\mathbf{o}}_t)P(\bar{\mathbf{o}}_t|\mathbf{s}_t, \bar{\mathbf{o}}_{t-1}), \tag{26}$$

Notice that the *Markovian master policy* $P(\bar{\mathbf{o}}_t|\mathbf{s}_t, \bar{\mathbf{o}}_{t-1})$ is actually the marginalization over the termination variable $\mathbf{b}_t$ in Eq. 24: $P(\bar{\mathbf{o}}_t|\mathbf{s}_t, \bar{\mathbf{o}}_{t-1}) = \sum_{\mathbf{b}_t} P(\mathbf{b}_t|\mathbf{s}_t, \bar{\mathbf{o}}_{t-1})P(\bar{\mathbf{o}}_t|\mathbf{b}_t, \mathbf{s}_t, \bar{\mathbf{o}}_{t-1})$. The joint distribution is factorized from Eq. 25 to Eq. 26 by following conditional independences in the PGM (Figure 17):

We define a linear function $\bar{f}(\bar{\mathbf{o}}) = \bar{\mathbf{o}} \cdot \mathbf{d}^T : \bar{\mathbb{O}} \to \mathbb{O}$ which maps $\bar{\mathbf{o}}$ to $\mathbf{o}$, where $\mathbf{d} = [1, 2, \ldots, K]^T$ is a $K$-dimensional constant integer vector, and hence $\bar{f}(\bar{\mathbf{o}}) = \mathbf{o}$. Note that $\bar{f}$ is a *Bijection* since it is a linear function defined on a finite integer space. We define a tuple of partition function $\bar{B} = <\bar{f}, \bar{I}, \bar{I}, \bar{f}>$ where $\bar{I}$ is the identical function and is also a Bijection. Therefore, the partition function $\bar{B}$ is a tuple of Bijection functions $\bar{B}(\bar{\mathbf{s}}, \bar{\mathbf{a}}) = \bar{B}(\{\bar{\mathbf{o}}_{t-1}, \mathbf{s}_t, \mathbf{a}_t, \bar{\mathbf{o}}_t\}) = \{\mathbf{o}_{t-1}, \mathbf{s}_t, \mathbf{a}_t, \mathbf{o}_t\}$, which maps $\bar{\tau}$ to $\tau$, where $\bar{\tau} = \{\mathbf{s}_0, \bar{\mathbf{o}}_0, \mathbf{a}_0, \mathbf{s}_1, \bar{\mathbf{o}}_1, \mathbf{a}_1, \ldots\}$ is the trajectory of the *HiT-MDP*. Therefore,

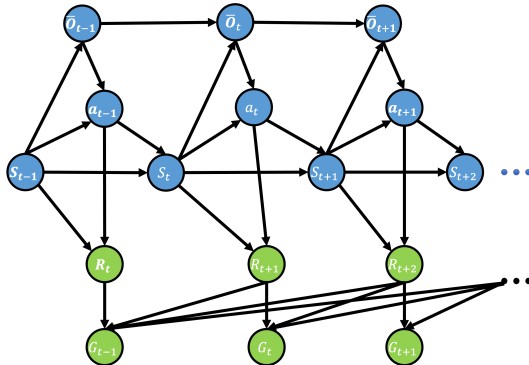

**Figure 17:** PGM of the *HiT-MDP*

by following the *Partition Function* $\bar{B}$, the dynamics $P(\tau)$ of the *SMDP-Option* in Eq. 24 under the Bijection $\bar{B}$ is equivalent to the dynamics $P(\bar{\tau})$ of the *HiT-MDP* $P(\tau/\bar{B}) \equiv P(\bar{\tau}/\bar{B})$.

As for the reward function $r$, notice that from the PGM we have the conditional independence $\mathbf{r}_{t+1} \perp\!\!\!\perp \bar{\mathbf{o}}_t | \mathbf{a}_t$. Therefore, *SMDP-Option* and the *HiT-MDP* also share the same expected reward function $r_{t+1}(\mathbf{s}_t, \mathbf{a}_t, \mathbf{o}_t) = E[\mathbf{r}_{t+1}|\mathbf{s}_t, \mathbf{a}_t, \mathbf{o}_t] = E[\mathbf{r}_{t+1}|\mathbf{s}_t, \mathbf{a}_t] = r_{t+1}(\mathbf{s}_t, \mathbf{a}_t)$. Therefore, the SMDP-based option framework has an MDP-based equivalence:

**Theorem C.1.** *By the definition of Bisimulation Relation, the SMDP-based option framework, which employs Markovian options, has an underlying MDP equivalence because:*

1. $$P(\tau/\bar{B}) \equiv P(\bar{\tau}/\bar{B}) \text{ and } \bar{B} \text{ is a Bijection.}$$

2. $$r(\tau/\bar{B}) \equiv r(\bar{\tau}/\bar{B}).$$

Although the reward function $r$ is equivalent, in Section D we identify a problem that under the configuration of HiT-MDPs, the conventional value function $V[\mathbf{s}_t]$ does not yield a Bellman equation. We have to propose a *Markovian Option-Value Function* $V[\mathbf{s}_t, \bar{\mathbf{o}}_{t-1}]$ to solve this problem and prove that it is equivalent to $V[\mathbf{s}_t]$. This is non-trivial since compared to the *SMDP-Option*, the MDP formulation introduces extra dependencies on $\bar{\mathbf{o}}$. In Section D.2, by exploiting conditional independencies we prove that they do have the same expected return under the Bijection $\bar{B}$.

## D  PROOF OF THE OPTIMAL VALUE EQUIVALENCE

In this Section we derive value functions of HiT-MDPs and prove its optimal value equivalence to the option framework. Following the structure of Section C, we first reformulate value functions of SMDP-Option into an MDP formulation by following the dynamics in Section C.2. We then following the HiT-MDPs' dynamics proposed in Section C.3 and derive a *Markovian Option-Value function* of HiT-MDPs, and prove it converges to the optimal value. In the end we prove that MDP-Option is optimal value equivalent to the SMDP-Option.

### D.1  THE MDP FORMULATED OPTION FRAMEWORK'S VALUE FUNCTIONS

With dynamics proposed in C.2, we now derive the MDP formulated value functions for the option framework (Bacon et al., 2017; Sutton et al., 1999). We follow Sutton & Barto (2018)'s notation in this section and write value functions for MDP below:

$$V[\mathbf{s}_t] = \mathbb{E}[G_t|\mathbf{s}_t] = \sum_{G_t} G_t \sum_{\mathbf{o}_t} P(G_t, \mathbf{o}_t|\mathbf{s}_t)$$

$$= \sum_{\mathbf{o}_t} P(\mathbf{o}_t|\mathbf{s}_t) \sum_{G_t} G_t P(G_t|\mathbf{s}_t, \mathbf{o}_t)$$

$$= \sum_{\mathbf{o}_t} P(\mathbf{o}_t|\mathbf{s}_t) \mathbb{E}[G_t|\mathbf{o}_t, \mathbf{s}_t]$$

$$= \sum_{\mathbf{o}_t} P(\mathbf{o}_t|\mathbf{s}_t) Q_O[\mathbf{o}_t, \mathbf{s}_t], \tag{27}$$

where $V[\mathbf{s}_t]$ is the state value function(Sutton & Barto, 2018) and $Q_O[\mathbf{o}_t, \mathbf{s}_t]$ is the option value function(Bacon et al., 2017; Sutton et al., 1999), $G_t = \sum_t r_{t+1}$ is the return. Note that in deriving equation (27) we only use summation rule and production rule, the conditional dependency relationships in PGM (figure 16) are not used.

The option value function $Q_O[\mathbf{o}_t, \mathbf{s}_t]$ can be further expanded as:

$$Q_O[\mathbf{o}_t, \mathbf{s}_t] = \mathbb{E}[G_t|\mathbf{o}_t, \mathbf{s}_t] = \sum_{\mathbf{a}_t} P(\mathbf{a}_t|\mathbf{s}_t, \mathbf{o}_t)\mathbb{E}[G_t|\mathbf{o}_t, \mathbf{s}_t, \mathbf{a}_t]$$

$$= \sum_{\mathbf{a}_t} P(\mathbf{a}_t|\mathbf{s}_t, \mathbf{o}_t) Q_U[\mathbf{o}_t, \mathbf{s}_t, \mathbf{a}_t], \tag{28}$$

where $Q_U[\mathbf{o}_t, \mathbf{s}_t, \mathbf{a}_t]$ is the option-action value function.

**Proposition D.1.1.** *MDP formulation has identical state value function $V[\mathbf{s}_t]$ and option value function $Q_O[\mathbf{o}_t, \mathbf{s}_t]$ to SMDP formulations*

*Proof.* Note that in derivations above we only use summation and production rules. Both equation (27) and (28) are identical to the conventional SMDP option framework. □

From now on, we will continue derivations with conditonal independence relationships encoded in PGM (Chapter 8.2.1 (Bishop, 2006)). We have following conditional independence relationships from PGM (figure 16):

$$\{R_{t+2}, G_{t+1}\} \perp\!\!\!\perp \{\mathbf{b}_{t+1}\} \qquad\qquad | \{\mathbf{o}_{t+1}\}, \tag{29}$$
$$\{R_{t+2}, G_{t+1}\} \perp\!\!\!\perp \{\mathbf{s}_t\} \qquad\qquad | \{\mathbf{s}_{t+1}, \mathbf{o}_t\}, \tag{30}$$
$$\{R_{t+2}, G_{t+1}\} \perp\!\!\!\perp \{\mathbf{a}_t\} \qquad\qquad | \{\mathbf{s}_{t+1}\}, \tag{31}$$
$$\{R_{t+2}, G_{t+1}\} \perp\!\!\!\perp \{\mathbf{o}_t\} \qquad\qquad | \{\mathbf{s}_{t+1}, \mathbf{o}_{t+1}\}, \tag{32}$$
$$\{R_{t+1}, G_t, \mathbf{s}_{t+1}\} \perp\!\!\!\perp \{\mathbf{o}_t\} \qquad\qquad | \{\mathbf{a}_t\}. \tag{33}$$

With above conditional independence relationships in hand, we now show that the MDP formulation has identical value functions to the conventional SMDP formulation(Sutton et al., 1999; Bacon et al., 2017).

**Proposition D.1.2.** *MDP formulation has identical option-action value function $Q_U[\mathbf{o}_t, \mathbf{s}_t, \mathbf{a}_t]$ to SMDP formulations*

$$Q_U[\mathbf{o}_t, \mathbf{s}_t, \mathbf{a}_t] = r(\mathbf{s}_t, \mathbf{a}_t) + \gamma \sum_{\mathbf{s}_{t+1}} P(\mathbf{s}_{t+1}|\mathbf{s}_t, \mathbf{a}_t) U[\mathbf{s}_{t+1}, \mathbf{o}_t]. \tag{34}$$

*Proof.*

$$
\begin{aligned}
Q_U[\mathbf{o}_t, \mathbf{s}_t, \mathbf{a}_t] =& \mathbb{E}[G_t | \mathbf{o}_t, \mathbf{s}_t, \mathbf{a}_t] \\
=& \mathbb{E}[R_{t+1} + \gamma G_{t+1} | \mathbf{o}_t, \mathbf{s}_t, \mathbf{a}_t] && \text{by definition of } G_t \\
=& \mathbb{E}[R_{t+1} | \mathbf{s}_t, \mathbf{a}_t] + && \text{use eq (33)} \\
& \gamma \sum_{G_{t+1}} G_{t+1} \sum_{\mathbf{s}_{t+1}} P(\mathbf{s}_{t+1} | \mathbf{s}_t, \mathbf{o}_t, \mathbf{a}_t) P(G_{t+1} | \mathbf{s}_{t+1}, \mathbf{o}_t, \mathbf{s}_t, \mathbf{a}_t) \\
=& r(\mathbf{s}_t, \mathbf{a}_t) + \\
& \gamma \sum_{G_{t+1}} G_{t+1} \sum_{\mathbf{s}_{t+1}} P(\mathbf{s}_{t+1} | \mathbf{s}_t, \mathbf{a}_t) P(G_{t+1} | \mathbf{s}_{t+1}, \mathbf{o}_t) && \text{use eq 30 31 and 33} \\
=& r(\mathbf{s}_t, \mathbf{a}_t) + \gamma \sum_{\mathbf{s}_{t+1}} P(\mathbf{s}_{t+1} | \mathbf{s}_t, \mathbf{a}_t) \mathbb{E}[G_{t+1} | \mathbf{s}_{t+1}, \mathbf{o}_t] \\
=& r(\mathbf{s}_t, \mathbf{a}_t) + \gamma \sum_{\mathbf{s}_{t+1}} P(\mathbf{s}_{t+1} | \mathbf{s}_t, \mathbf{a}_t) U[\mathbf{s}_{t+1}, \mathbf{o}_t].
\end{aligned}
$$

$\square$

However, unlike conventional value functions $V[\mathbf{s}_t]$, $Q_O[\mathbf{o}_t, \mathbf{s}_t]$, $Q_U[\mathbf{o}_t, \mathbf{s}_t, \mathbf{a}_t]$ of the option framework derived in Section D.1, the value function $U[\mathbf{s}_{t+1}, \bar{\mathbf{o}}_t]$ has an extra dependency on the hidden variable $\bar{\mathbf{o}}_t$. Therefore, conventional value function $V[\mathbf{s}_t]$ does not yield a Bellman equation under the configuration of HiT-MDPs. To tackle this issue, we propose the *Markovian Option-Value Function* $\bar{V}$ (derivations of Eq. (6) in the main text):

$$
\begin{aligned}
\bar{V}[\mathbf{s}_t, \bar{\mathbf{o}}_{t-1}] =& \mathbb{E}[G_t | \mathbf{s}_t, \bar{\mathbf{o}}_{t-1}] \\
=& \sum_{\bar{\mathbf{o}}_t} P(\bar{\mathbf{o}}_t | \mathbf{s}_t, \bar{\mathbf{o}}_{t-1}) \mathbb{E}(G_t | \mathbf{s}_t, \bar{\mathbf{o}}_t, \bar{\mathbf{o}}_{t-1}) \\
=& \sum_{\bar{\mathbf{o}}_t} P(\bar{\mathbf{o}}_t | \mathbf{s}_t, \bar{\mathbf{o}}_{t-1}) \mathbb{E}[G_t | \mathbf{s}_t, \bar{\mathbf{o}}_t] \\
=& \sum_{\bar{\mathbf{o}}_t} P(\bar{\mathbf{o}}_t | \mathbf{s}_t, \bar{\mathbf{o}}_{t-1}) Q_O[\bar{\mathbf{o}}_t, \mathbf{s}_t], && (35)
\end{aligned}
$$

where from line 2 to line 3 we use the conditional independence property in PGM that $G_t \perp\!\!\!\perp \bar{\mathbf{o}}_{t-1} | \{\mathbf{s}_t, \bar{\mathbf{o}}_t\}$.

**Proposition D.1.3.** *MDP formulation has identical option-value function upon arrival $U[\mathbf{s}_{t+1}, \mathbf{o}_t]$ to SMDP formulations*[5]

$$
\begin{aligned}
U[\mathbf{s}_{t+1}, \mathbf{o}_t] =& (1 - \beta_{t+1}) Q_O[\mathbf{o}_{t+1} = \mathbf{o}_t, \mathbf{s}_{t+1}] + \beta_{t+1} V[\mathbf{s}_{t+1}] && (36) \\
=& Q_O[\mathbf{o}_{t+1} = \mathbf{o}_t, \mathbf{s}_{t+1}] - \beta_{t+1} A[\mathbf{o}_{t+1} = \mathbf{o}_t, \mathbf{s}_{t+1}]. && (37)
\end{aligned}
$$

---

[5]Both equations (36) and (37) is largely used in the conventional SMDP papers(Sutton et al., 1999; Bacon et al., 2017).

*Proof.*

$$U[\mathbf{s}_{t+1}, \mathbf{o}_t] = \mathbb{E}[G_{t+1}|\mathbf{s}_{t+1}, \mathbf{o}_t]$$

$$= \sum_{G_{t+1}} G_{t+1}$$

$$\sum_{\mathbf{o}_{t+1}} \sum_{\mathbf{b}_{t+1}} P(\mathbf{b}_{t+1}|\mathbf{o}_t, \mathbf{s}_{t+1}) P(\mathbf{o}_{t+1}|\mathbf{b}_{t+1}, \mathbf{o}_t, \mathbf{s}_{t+1}) P(G_{t+1}|\mathbf{o}_{t+1}, \mathbf{b}_{t+1}, \mathbf{o}_t, \mathbf{s}_{t+1})$$

$$= \sum_{\mathbf{o}_{t+1}} \sum_{\mathbf{b}_{t+1}} P(\mathbf{b}_{t+1}|\mathbf{o}_t, \mathbf{s}_{t+1}) P(\mathbf{o}_{t+1}|\mathbf{b}_{t+1}, \mathbf{o}_t, \mathbf{s}_{t+1}) \sum_{G_{t+1}} G_{t+1} P(G_{t+1}|\mathbf{o}_{t+1}, \mathbf{s}_{t+1})$$

$$= \sum_{\mathbf{o}_{t+1}} \left[ (1 - \beta_{t+1})\mathbf{1}_{\mathbf{o}_{t+1}=\mathbf{o}_t} + \beta_{t+1}P(\mathbf{o}_{t+1}|\mathbf{s}_{t+1}) \right] Q_O[\mathbf{o}_{t+1}, \mathbf{s}_{t+1}]$$

$$= (1 - \beta_{t+1})Q_O[\mathbf{o}_{t+1} = \mathbf{o}_t, \mathbf{s}_{t+1}] + \beta_{t+1}V[\mathbf{s}_{t+1}]$$

$$= Q_O[\mathbf{o}_{t+1} = \mathbf{o}_t, \mathbf{s}_{t+1}] - \beta_{t+1}A[\mathbf{o}_{t+1} = \mathbf{o}_t, \mathbf{s}_{t+1}].$$

from line 3 to line 4 use equation (29) and (32). From line 4 to line 5 use equation (22) and definition of $Q_O$. The second last line use equation (27). The last line use the definition of advantage function $A$. □

Under our MDP formulation, we also propose proposition D.1.4. We derive our gradient theorems based on equation (38) in section E. This important relationship largely simplify derivations than the original paper (Bacon et al., 2017) as well as give rise to the the HiT-MDP.

**Proposition D.1.4.** *The option-value function upon arrival* $U[\mathbf{s}_{t+1}, \mathbf{o}_t]$ *is an expectation over option value function* $Q_O[\mathbf{o}_{t+1}, \mathbf{s}_{t+1}]$ *conditioned on previous option* $O_t$

$$U[\mathbf{s}_{t+1}, \mathbf{o}_t] = \sum_{\mathbf{o}_{t+1}} P(\mathbf{o}_{t+1}|\mathbf{o}_t, \mathbf{s}_{t+1}) Q_O[\mathbf{o}_{t+1}, \mathbf{s}_{t+1}]. \tag{38}$$

*Proof.* Following proof of proposition D.1.3,

$$U[\mathbf{s}_{t+1}, \mathbf{o}_t] = \sum_{\mathbf{o}_{t+1}} \sum_{\mathbf{b}_{t+1}} P(\mathbf{b}_{t+1}|\mathbf{o}_t, \mathbf{s}_{t+1}) P(\mathbf{o}_{t+1}|\mathbf{b}_{t+1}, \mathbf{o}_t, \mathbf{s}_{t+1}) \sum_{G_{t+1}} G_{t+1} P(G_{t+1}|\mathbf{o}_{t+1}, \mathbf{s}_{t+1})$$

$$= \sum_{\mathbf{o}_{t+1}} P(\mathbf{o}_{t+1}|\mathbf{o}_t, \mathbf{s}_{t+1}) Q_O[\mathbf{o}_{t+1}, \mathbf{s}_{t+1}].$$

□

## D.2 THE OPTIMAL VALUE EQUIVALENCE BETWEEN HiT-MDP AND OPTION FRAMEWORKS

**Theorem D.1.** *The option-action value function Eq. 7 satisfies the Bellman Operator* $\mathcal{T}^{\mathcal{H}}$

$$\mathcal{T}^{\mathcal{H}}Q_A[\mathbf{s}_t, \bar{\mathbf{o}}_t, \mathbf{a}_t] = \mathbb{E}[G_t|\mathbf{s}_t, \bar{\mathbf{o}}_t, \mathbf{a}_t]$$

$$= r(s, a) + \gamma \sum_{\mathbf{s}_{t+1}} P(\mathbf{s}_{t+1}|\mathbf{s}_t, \mathbf{a}_t)\bar{V}[\mathbf{s}_{t+1}, \bar{\mathbf{o}}_t], \tag{39}$$

*where the Markovian option-value function given by Eq. 6.*

*Proof.* Following dynamics derived in Section C.3, we can define value functions on HiT-MDPs as (derivations of Eq. (8) in the main text):

$$Q_A[\mathbf{s}_t, \bar{\mathbf{o}}_t, \mathbf{a}_t] = \mathbb{E}[G_t|\mathbf{s}_t, \bar{\mathbf{o}}_t, \mathbf{a}_t] = \mathbb{E}[R_{t+1} + \gamma G_{t+1}|\mathbf{s}_t, \bar{\mathbf{o}}_t, \mathbf{a}_t]$$

$$= r(s, o, a) + \gamma \sum_{\mathbf{s}_{t+1}} P(\mathbf{s}_{t+1}|\mathbf{s}_t, \bar{\mathbf{o}}_t, \mathbf{a}_t)\mathbb{E}[G_{t+1}|\mathbf{s}_{t+1}, \mathbf{s}_t, \bar{\mathbf{o}}_t, \mathbf{a}_t]$$

$$= r(s, a) + \gamma \sum_{\mathbf{s}_{t+1}} P(\mathbf{s}_{t+1}|\mathbf{s}_t, \mathbf{a}_t)\mathbb{E}[G_{t+1}|\mathbf{s}_{t+1}, \bar{\mathbf{o}}_t]$$

$$= r(s, a) + \gamma \sum_{\mathbf{s}_{t+1}} P(\mathbf{s}_{t+1}|\mathbf{s}_t, \mathbf{a}_t)\bar{V}[\mathbf{s}_{t+1}, \bar{\mathbf{o}}_t],$$

where from line 2 to line 3 we use the conditional independence property in PGM that $R_{t+1} \perp\!\!\!\perp \bar{\mathbf{o}}_t | \mathbf{a}_t$, $G_{t+1} \perp\!\!\!\perp \mathbf{s}_t | \{\mathbf{s}_{t+1}, \bar{\mathbf{o}}_t\}$ and $G_{t+1} \perp\!\!\!\perp \mathbf{a}_t | \mathbf{s}_{t+1}$. $\gamma \in \mathbb{R}$ is a discounting factor. The last line uses the definition of the *Markovian option value function* (Eq. 35). $\qquad\square$

**Theorem D.2.** *(Markovian Option Policy Evaluation Theorem). Assume that throughout our computation the $Q_A[\cdot, \cdot]$ and $\bar{V}[\cdot]$ are bounded and $\mathbb{A} < \infty$, the sequence $Q_A^k$ defined by $Q_A^{k+1} = \mathcal{T}^{\mathcal{H}} Q_A^k$ will converge to the option-action value function $Q_A^{\pi_A}$ as $k \to \infty$.*

*Proof.* As with the standard convergence results for policy evaluation (Sutton & Barto, 2018), by the definition of $\mathcal{T}^{\mathcal{H}}$ (Eq. 8) the *option-action value function* $Q_A^{\pi_A}$ is a fixed point.

To prove the $\mathcal{T}^H$ is a contraction, define a norm on $V$-values functions $V$ and $U$

$$\|V - U\|_\infty \triangleq \max_{\bar{s} \in \bar{S}} |V(\bar{s}) - U(\bar{s})|. \tag{40}$$

where $\bar{s} = \{s, o\}$.

By recurssively apply the Hidden Temporal Bellman Operator $\mathcal{T}^H$, we have:

$$\bar{V}[\mathbf{s}_t, \bar{\mathbf{o}}_{t-1}] = \mathbb{E}[G_t | \mathbf{s}_t, \bar{\mathbf{o}}_{t-1}] = \sum_{\bar{\mathbf{o}}_t} P(\bar{\mathbf{o}}_t | \mathbf{s}_t, \bar{\mathbf{o}}_{t-1}) Q_O[\mathbf{s}_t, \bar{\mathbf{o}}_t]$$

$$= \sum_{\bar{\mathbf{o}}_t} P(\bar{\mathbf{o}}_t | \mathbf{s}_t, \bar{\mathbf{o}}_{t-1}) \sum_{\mathbf{a}_t} P(\mathbf{a}_t | \mathbf{s}_t, \bar{\mathbf{o}}_t) \left[ r(s, a) + \gamma \sum_{\mathbf{s}_{t+1}} P(\mathbf{s}_{t+1} | \mathbf{s}_t, \mathbf{a}_t) \bar{V}[\mathbf{s}_{t+1}, \bar{\mathbf{o}}_t] \right]$$

$$= r(s, a) + \gamma \sum_{\bar{\mathbf{o}}_t} P(\bar{\mathbf{o}}_t | \mathbf{s}_t, \bar{\mathbf{o}}_{t-1}) \sum_{\mathbf{a}_t} P(\mathbf{a}_t | \mathbf{s}_t, \bar{\mathbf{o}}_t) \sum_{\mathbf{s}_{t+1}} P(\mathbf{s}_{t+1} | \mathbf{s}_t, \mathbf{a}_t) \bar{V}[\mathbf{s}_{t+1}, \bar{\mathbf{o}}_t]$$

$$= r(s, a) + \gamma \sum_{\bar{\mathbf{o}}_t, \mathbf{s}_{t+1}} P(\mathbf{s}_{t+1}, \bar{\mathbf{o}}_t | \mathbf{s}_t, \bar{\mathbf{o}}_{t-1}) \bar{V}[\mathbf{s}_{t+1}, \bar{\mathbf{o}}_t]$$

$$= r(s, a) + \gamma \mathbb{E}_{\mathbf{s}_{t+1}, \bar{\mathbf{o}}_t} \left[ \bar{V}[\mathbf{s}_{t+1}, \bar{\mathbf{o}}_t] \right] \tag{41}$$

Therefore, by applying Eq. 41 to $V$ and $U$ we have:

$$\|T^\pi V - T^\pi U\|_\infty$$

$$= \max_{\bar{s} \in \bar{S}} \left| \gamma \mathbb{E}_{\mathbf{s}_{t+1}, \bar{\mathbf{o}}_t} \left[ \bar{V}[\mathbf{s}_{t+1}, \bar{\mathbf{o}}_t] \right] - \gamma \mathbb{E}_{\mathbf{s}_{t+1}, \bar{\mathbf{o}}_t} \left[ \bar{U}[\mathbf{s}_{t+1}, \bar{\mathbf{o}}_t] \right] \right|$$

$$= \gamma \max_{\bar{s} \in \bar{S}} \mathbb{E}_{\mathbf{s}_{t+1}, \bar{\mathbf{o}}_t} \left[ \left| \bar{V}[\mathbf{s}_{t+1}, \bar{\mathbf{o}}_t] - \bar{U}[\mathbf{s}_{t+1}, \bar{\mathbf{o}}_t] \right| \right]$$

$$\leq \gamma \max_{\bar{s} \in \bar{S}} \mathbb{E}_{\mathbf{s}_{t+1}, \bar{\mathbf{o}}_t} \left[ \gamma \max_{\bar{s} \in \bar{S}} \left| \bar{V}[\mathbf{s}_{t+1}, \bar{\mathbf{o}}_t] - \bar{U}[\mathbf{s}_{t+1}, \bar{\mathbf{o}}_t] \right| \right]$$

$$\leq \gamma \max_{\bar{s} \in \bar{S}} |V[\bar{s}] - U[\bar{s}]|$$

$$= \gamma \|V - U\|_\infty \tag{42}$$

Therefore, $\mathcal{T}^{\mathcal{H}}$ is a contraction and by the fixed point theorem, Theorem 3.3 follows immediately. $\square$

**Proposition D.2.1.** *The option-induced* Markovian Option-Value Function $\bar{V}$ *is equivalent to the conventional value function $V$*

*Proof.* for Proposition 3.4: By law of total expectation:

$$\mathbb{E}_{\bar{\mathbf{o}}_{t-1}}[V[\mathbf{s}_t, \bar{\mathbf{o}}_{t-1}]] = \mathbb{E}_{\bar{\mathbf{o}}_{t-1}}[\mathbb{E}[G_t | \mathbf{s}_t, \bar{\mathbf{o}}_{t-1}]] = \mathbb{E}[G_t | \mathbf{s}_t] = V[\mathbf{s}_t]$$

thus $V[\mathbf{s}_t, \bar{\mathbf{o}}_{t-1}]$ is an unbiased estimator of $V[\mathbf{s}_t]$, with conditional independences defined in PGM 17. $\qquad\square$

**Proposition D.2.2.** *The option-induced* Markovian Option-Value Function $\bar{V}$ *has smaller variance than the conventional value function $V$*

*Proof.* for Proposition 3.5: By law of total conditional variance:

$$\begin{aligned}
\mathrm{Var}(V[\mathbf{s}_t]) = \mathrm{Var}([\mathbb{E}[G_t|\mathbf{s}_t]]) &= \mathbb{E}[\mathrm{Var}(\mathbb{E}[G_t|\mathbf{s}_t,\bar{\mathbf{o}}_{t-1}])|\mathbf{s}_t] + \mathrm{Var}(\mathbb{E}[\mathbb{E}[G_t|\mathbf{s}_t,\bar{\mathbf{o}}_{t-1}]]|\mathbf{s}_t) \\
&= \mathbb{E}[\mathrm{Var}(V[\mathbf{s}_t,\bar{\mathbf{o}}_{t-1}])|\mathbf{s}_t] + \mathrm{Var}(\mathbb{E}[V[\mathbf{s}_t,\bar{\mathbf{o}}_{t-1}]]|\mathbf{s}_t) \\
&\geq \mathrm{Var}(\mathbb{E}[V[\mathbf{s}_t,\bar{\mathbf{o}}_{t-1}]]|\mathbf{s}_t),
\end{aligned}$$

with conditional independences defined in PGM 17 $\qquad\square$

## E    MAX-ENTROPY OPTION POLICY GRADIENTS

In the main text Section 4 we solve the HiT-MDPs under the maximum entropy framework (Levine, 2018). Here we proof the Evidence Lower BOund (ELBO) proposed in the main text and derive the Maximum entropy Option Policy Gradient (MOPG) theorems.

### E.1    THE MAXIMUM ENTROPY OBJECTIVE FUNCTION

Following notations defined in the main text Section 4, optimality variables $e_{1:T}$ ($e$ denotes the realization of $\mathbf{e}$ when $\mathbf{e} = 1$) are treated as observed variables while random variables from the trajectory $\bar{\tau} = \{\mathbf{s}, \mathbf{a}, \bar{\mathbf{o}}, ...\}$ are treated as latent variables. The variational Evidence Lower BOund (ELBO) is given by:

**Theorem E.1.** *The problem of learning optimal action and master policies can be simplified as shrinking the KL-Divergence:* $D_{KL}[P(\bar{\tau})||q(\bar{\tau}|e_{1:T})]$

*Proof.*

$$\begin{aligned}
\log q(\mathbf{e}_{1:T}) &= \log \int\int\int q(\mathbf{e}_{1:T},\mathbf{s}_{1:T},\mathbf{a}_{1:T},\bar{\mathbf{o}}_{1:T})d\mathbf{s}_{1:T}d\mathbf{a}_{1:T}d\bar{\mathbf{o}}_{1:T} \\
&= \log \int\int\int q(\mathbf{e}_{1:T},\mathbf{s}_{1:T},\mathbf{a}_{1:T},\bar{\mathbf{o}}_{1:T})\frac{P(\bar{\tau})}{P(\bar{\tau})}d\mathbf{s}_{1:T}d\mathbf{a}_{1:T}d\bar{\mathbf{o}}_{1:T} \\
&= \log E_{(\mathbf{s}_{1:T},\mathbf{a}_{1:T},\bar{\mathbf{o}}_{1:T})\sim P(\bar{\tau})}\left[\frac{q(\mathbf{s}_{1:T},\mathbf{a}_{1:T},\bar{\mathbf{o}}_{1:T}|e_{1:T})q(e_{1:T})}{P(\bar{\tau})}\right] \quad (43) \\
&\geq E_{(\mathbf{s}_{1:T},\mathbf{a}_{1:T},\bar{\mathbf{o}}_{1:T})\sim P(\bar{\tau})}\left[\log q(\mathbf{s}_{1:T},\mathbf{a}_{1:T},\bar{\mathbf{o}}_{1:T}|e_{1:T}) - \log P(\bar{\tau})\right] \\
&= -D_{\mathrm{KL}}[P(\bar{\tau})||q(\bar{\tau}|e_{1:T})]
\end{aligned}$$

where third last line to the second last line follows from the Jensen's inequality and $P(\bar{\tau})$ is the dynamics of the HiT-MDPs defined in Eq. 26 $\qquad\square$

**Theorem E.2.** *(Markovian Option Policy Iteration Theorem). Repeated application of Markovian Option evaluation and improvement to any* $\pi_{O,A} \in \prod$ *converges to a policy* $\pi_{O,A}^*$ *such that* $Q^{\pi_{O,A}^*}(\bar{\mathbf{s}}_t,\bar{\mathbf{a}}_t) \geq Q^{\pi_{O,A}}(\bar{\mathbf{s}}_t,\bar{\mathbf{a}}_t)$ *for all* $\pi_{O,A} \in \prod$ *and* $\bar{\mathbf{s}}_t,\bar{\mathbf{a}}_t \in \bar{\mathbb{S}} \times \bar{\mathbb{A}}$, *assuming* $|\bar{\mathbb{A}}| < \infty$.

*Proof.* In this proof we use $\pi_A$ for illustration, the proof of $\pi_O$ follows the same derivation. From Theorem 4.1 we have that:

$$\begin{aligned}
\pi_A^* &= \arg\max_{\pi_A} -D_{\mathrm{KL}}[P(\bar{\tau})||q(\bar{\tau}|e_{1:T})] \\
&= \arg\max_{\pi_A} \sum_t E_{\pi_A}[r(\mathbf{s}_t,\mathbf{a}_t) + I[\mathbf{o}_t] + \mathcal{H}(\pi_O) + \mathcal{H}(\pi_A)], \\
&= \arg\max_{\pi_A} \sum_t E_{\pi_A}[r(\mathbf{s}_t,\mathbf{a}_t) + I[\mathbf{o}_t] + \mathcal{H}(\pi_A)], \quad (44) \\
&= \arg\max_{\pi_A} \sum_t E_{\pi_A}[r(\mathbf{s}_t,\mathbf{a}_t) + I[\mathbf{o}_t] + \mathcal{H}(\pi_A)],
\end{aligned}$$

Therefore, it must be the case that $E_{\mathbf{a}_t\sim\pi_A^{new}}[r(\mathbf{s}_t,\mathbf{a}_t) + I(\mathbf{o}_t) + \mathcal{H}(\pi_A^{old})] \geq E_{\mathbf{a}_t\sim\pi_A^{old}}[r(\mathbf{s}_t,\mathbf{a}_t) + I[\mathbf{o}_t] + \mathcal{H}(\pi_A^{old})] = \bar{V}^{\pi_A^{old}}[s,\bar{o}]$, since we can always choose $\pi_A^{new} = \pi_A^{old} \in \prod$. Substituting this

inequality into Theorem 3.2 leads to:

$$
\begin{aligned}
Q_A^{\pi_A^{old}}[\mathbf{s}, \bar{\mathbf{o}}, \mathbf{a}] &= r(\mathbf{s}, \mathbf{a}) + I[\mathbf{o}] + \gamma E_{\mathbf{s}}[\bar{V}^{\pi_A^{old}}[\boldsymbol{s}, \bar{\mathbf{o}}]] \\
&\leq r(\mathbf{s}, \mathbf{a}) + I[\mathbf{o}] + \gamma E_{\mathbf{s}}[E_{\mathbf{a} \sim \pi_A^{new}}[r(\mathbf{s}_{t+1}, \mathbf{a}_{t+1}) + I[\mathbf{o}_{t+1}] + \mathcal{H}(\pi_A^{old})]] \qquad (45) \\
&\leq Q_A^{\pi_A^{new}}[\mathbf{s}, \bar{\mathbf{o}}, \mathbf{a}]
\end{aligned}
$$

where the convergence to $Q_A^{\pi_A^{new}}$ follows from the Markovian Option Policy Evaluation Theorem D.2. Therefore, the iteration of the sequence $Q_A^{\pi_A^i}$ is monotonically increasing and we get $Q_A^{\pi_A^*} > Q_A^{\pi_A}$ for all $\pi_A \neq \pi_A^*$ and $(\mathbf{s}, \mathbf{o}, \mathbf{a}) \in \mathbb{S} \times \mathbb{O} \times \mathbb{A}$. $\qquad \square$

## E.2 PROOF FOR THE MASTER POLICY GRADIENT THEOREM

*Proof.*

$$
\begin{aligned}
\frac{\partial Q_O[\mathbf{s}_t, \bar{\mathbf{o}}_t]}{\partial \theta_o} &= \sum_{\mathbf{a}_t} P(\mathbf{a}_t | \mathbf{s}_t, \bar{\mathbf{o}}_t) \Big[ r(s, a) + I(\bar{\mathbf{o}}'|\mathbf{s}, \mathbf{a}, \bar{\mathbf{o}}) + \mathcal{H}(\pi_{\theta_{\bar{o}}}^O) + \gamma \sum_{\mathbf{s}_{t+1}} P(\mathbf{s}_{t+1} | \mathbf{s}_t, \mathbf{a}_t) \frac{\partial V[\mathbf{s}_{t+1}, \bar{\mathbf{o}}_t]}{\partial \theta_o} \Big] \\
&= \sum_{\mathbf{s}_{t+1}} \gamma P(\mathbf{s}_{t+1} | \mathbf{s}_t, \bar{\mathbf{o}}_t) \frac{\partial V[\mathbf{s}_{t+1}, \bar{\mathbf{o}}_t]}{\partial \theta_o} \\
\frac{\partial V[\mathbf{s}_t, \bar{\mathbf{o}}_{t-1}]}{\partial \theta_o} &= \sum_{\bar{\mathbf{o}}_t} \frac{\partial P(\bar{\mathbf{o}}_t | \mathbf{s}_t, \bar{\mathbf{o}}_{t-1})}{\partial \theta_o} Q_O[\mathbf{s}_t, \bar{\mathbf{o}}_t] + I(\bar{\mathbf{o}}'|\mathbf{s}, \mathbf{a}, \bar{\mathbf{o}}) + \mathcal{H}(\pi_{\theta_{\bar{o}}}^O) + \gamma \sum_{\bar{\mathbf{o}}_t} P(\bar{\mathbf{o}}_t | \mathbf{s}_t, \bar{\mathbf{o}}_{t-1}) \frac{Q_O[\mathbf{s}_t, \bar{\mathbf{o}}_t]}{\partial \theta_o} \\
&= \sum_{\bar{\mathbf{o}}_t} \frac{\partial P(\bar{\mathbf{o}}_t | \mathbf{s}_t, \bar{\mathbf{o}}_{t-1})}{\partial \theta_o} Q_O[\mathbf{s}_t, \bar{\mathbf{o}}_t] + I(\bar{\mathbf{o}}'|\mathbf{s}, \mathbf{a}, \bar{\mathbf{o}}) + \mathcal{H}(\pi_{\theta_{\bar{o}}}^O) + \gamma \sum_{\mathbf{s}_{t+1}, \bar{\mathbf{o}}_t} P(\mathbf{s}_{t+1}, \bar{\mathbf{o}}_t | \mathbf{s}_t, \bar{\mathbf{o}}_{t-1}) \frac{\partial V[\mathbf{s}_{t+1}, \bar{\mathbf{o}}_t]}{\partial \theta_o} \\
&= -\sum_{k=0}^{\infty} \sum_{\mathbf{s}_{t+k}, \bar{\mathbf{o}}_{t+k-1}} \\
&\qquad P_\gamma^{(k)}(\mathbf{s}_{t+k}, \bar{\mathbf{o}}_{t+k-1} | \mathbf{s}_t, \bar{\mathbf{o}}_{t-1}) \sum_{\bar{\mathbf{o}}_{t+k}} \frac{\partial P(\bar{\mathbf{o}}_{t+k} | \mathbf{s}_{t+k}, \bar{\mathbf{o}}_{t+k-1})}{\partial \theta_o} Q_O[\mathbf{s}_{t+k}, \bar{\mathbf{o}}_{t+k}] + I(\bar{\mathbf{o}}'|\mathbf{s}, \mathbf{a}, \bar{\mathbf{o}}) + \mathcal{H}(\pi_{\theta_{\bar{o}}}^O) \\
&= \mathbb{E}\Big[ \frac{\partial P(\mathbf{o}'|\mathbf{s}', \mathbf{o})}{\partial \theta_o} Q_O[\mathbf{s}', \mathbf{o}'] + I(\bar{\mathbf{o}}'|\mathbf{s}, \mathbf{a}, \bar{\mathbf{o}}) + \mathcal{H}(\pi_{\theta_{\bar{o}}}^O) \mid \mathbf{s}_t, \bar{\mathbf{o}}_{t-1} \Big].
\end{aligned}
$$

$\qquad \square$

## E.3 PROOF FOR THE ACTION POLICY GRADIENT THEOREM

*Proof.* Similar to the first equation above, continue expanding gradients of $\frac{\partial Q_O}{\partial \theta_a}$ by equations (6) (28) and (8):

$$
\begin{aligned}
\frac{\partial Q_O[\mathbf{s}_t, \bar{\mathbf{o}}_t]}{\partial \theta_a} &= \sum_{\mathbf{a}_t} \frac{\partial P(\mathbf{a}_t | \mathbf{s}_t, \bar{\mathbf{o}}_t)}{\partial \theta_a} Q_A[\mathbf{s}_t, \bar{\mathbf{o}}_t, \mathbf{a}_t] + \mathcal{H}(\pi_{\theta_a}^A) + \gamma \sum_{\mathbf{s}_{t+1}} P(\mathbf{s}_{t+1} | \mathbf{s}_t, \bar{\mathbf{o}}_t) \frac{\partial V[\mathbf{s}_{t+1}, \bar{\mathbf{o}}_t]}{\partial \theta_a} \\
&= \sum_{\mathbf{a}_t} \frac{\partial P(\mathbf{a}_t | \mathbf{s}_t, \bar{\mathbf{o}}_t)}{\partial \theta_a} Q_A[\mathbf{s}_t, \bar{\mathbf{o}}_t, \mathbf{a}_t] + \mathcal{H}(\pi_{\theta_a}^A) + \gamma \sum_{\mathbf{s}_{t+1}, \bar{\mathbf{o}}_{t+1}} P(\mathbf{s}_{t+1}, \bar{\mathbf{o}}_{t+1} | \mathbf{s}_t, \bar{\mathbf{o}}_t) \frac{\partial Q_O[\mathbf{s}_{t+1}, \bar{\mathbf{o}}_{t+1}]}{\partial \theta_a} \\
&= -\sum_{k=0}^{\infty} \sum_{\mathbf{s}_{t+k}, \bar{\mathbf{o}}_{t+k}} \\
&\qquad P_\gamma^{(k)}(\mathbf{s}_{t+k}, \bar{\mathbf{o}}_{t+k} | \mathbf{s}_t, \bar{\mathbf{o}}_t) \sum_{\mathbf{a}_{t+k}} \frac{\partial P(\mathbf{a}_{t+k} | \mathbf{s}_{t+k}, \bar{\mathbf{o}}_{t+k})}{\partial \theta_a} Q_A[\mathbf{s}_{t+k}, \bar{\mathbf{o}}_{t+k}, \mathbf{a}_{t+k}] + \mathcal{H}(\pi_{\theta_a}^A) \\
&= \mathbb{E}\Big[ \frac{\partial P(\mathbf{a}_{t+k} | \mathbf{s}_{t+k}, \bar{\mathbf{o}}_{t+k})}{\partial \theta_a} Q_A[\mathbf{s}_{t+k}, \bar{\mathbf{o}}_{t+k}, \mathbf{a}_{t+k}] + \mathcal{H}(\pi_{\theta_a}^A) \mid \mathbf{s}_t, \bar{\mathbf{o}}_t \Big].
\end{aligned}
$$

$\qquad \square$

## F LEARNING ALGORITHM FOR MOPG

---

**Algorithm 1:** The MOPG Algorithm

---

1 Initialize the option embedding matrix $\boldsymbol{W}_S$
2 Assign Initial State: $\mathbf{s}_t \leftarrow \mathbf{s}_0$
3 Assign Initial Option: $\hat{\mathbf{o}}_{t-1} \leftarrow \hat{\mathbf{o}}_0$
4
5 **while** *Converge* **do**
6      # Rollout trajectories and store in replay buffer
7      **repeat**
8          Retrieve the option context vector $\hat{\mathbf{o}}_{t-1} = \boldsymbol{W}_S^T \cdot \hat{\mathbf{o}}_{t-1}$
9          Sample $\hat{\mathbf{o}}_t \sim P(\hat{\mathbf{o}}_t|\mathbf{s}_t, \hat{\mathbf{o}}_{t-1})$
10          Retrieve the option context vector $\hat{\mathbf{o}}_t = \boldsymbol{W}_S^T \cdot \hat{\mathbf{o}}_t$
11          Sample $\mathbf{a}_t \sim P(\mathbf{a}_t|\mathbf{s}_t, \hat{\mathbf{o}}_t)$
12          Compute $Q_O[\mathbf{s}_t, \hat{\mathbf{o}}_t]$ and $V[\mathbf{s}_t, \hat{\mathbf{o}}_{t-1}]$
13          Take action $\mathbf{a}_t$ in $\mathbf{s}_t$, observe new state $\mathbf{s}_{t+1}$ and reward $R_{t+1}$
14      **until** *Rollout Length Reached*
15
16      # Compute Advantages for option & action policies
17      Assign $t$ reversely, from $RolloutLength - 1$ to 1
18      **repeat**
19          Compute option Advantage
         $A_t^O = R_{t+1} + \gamma(V[\mathbf{s}_{t+1}, \hat{\mathbf{o}}_t] - V[\mathbf{s}_t, \hat{\mathbf{o}}_{t-1}] + \Delta I + \Delta \mathcal{H}^O) + \gamma\lambda A_{t+1}^O$
20          Compute action Advantage
         $A_t^A = R_{t+1} + \gamma(Q_O[\mathbf{s}_{t+1}, \hat{\mathbf{o}}_{t+1}] - Q_O[\mathbf{s}_t, \hat{\mathbf{o}}_t] + \Delta \mathcal{H}^A) + \gamma\lambda A_{t+1}^A$
21      **until** *Rollout Length Reached*
22
23      # $\lambda$ is the GAE coefficient used in PPO.
24      # Optimize PPO Obj
25      **while** $i < PPO\ Optimization\ Epochs$ **do**
26          $\theta_o \leftarrow PPO(\frac{\partial P(\mathbf{o}'|\mathbf{s}', \mathbf{o})}{\partial \theta_o}, A^O)$
27          $\theta_a \leftarrow PPO(\frac{\partial P(\mathbf{a}|\mathbf{s}, \mathbf{o})}{\partial \theta_a}, A^A)$
28      **end**
29 **end**

---

## G IMPLEMENTATION DETAILS

### G.1 NEURAL NETWORK ARCHITECTURE

Interpretability is a key property to apply RL agents in real-world applications. Attention Option Critic (AOC) (Chunduru & Precup, 2020) first introduces the attention mechanism into options. In AOC, each option has a unique attention mask over state $\ddot{\mathbf{s}} = \ddot{\boldsymbol{W}}_o \odot \mathbf{s}$. Therefore, each option has a unique activation field (e.g., initiation set) over the observation space. Options learned by AOC are also interpretable since each option attends to different context in the state vector.

Our methods contributes to the option framework from a different aspect. In AOC, an option is still a tuple of $(\mathbb{I}, \beta, \pi_A)$ while in our work an option is an embedding vector $\hat{\mathbf{o}}$. The attention mechanism is used as a distance measure to compare which option is closer to the concatenated state-option pair (as shown in Eq. 48). As explained in the main text Section 5.3, options in HiT-MDP can be learned as option embeddings. Inspired by the *Transformer* (Vaswani et al., 2017), in this section we implement the

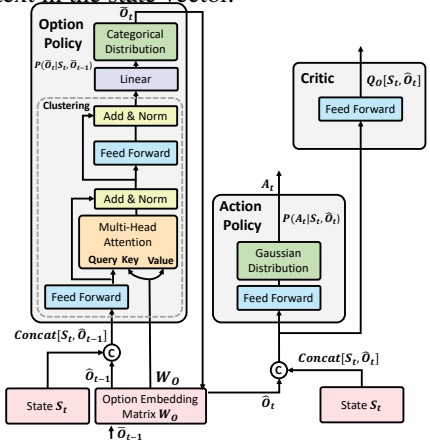

HiT-MDP as a simple yet effective Multi-Head Attention (MHA) (Vaswani et al., 2017) based Encoder-Decoder architecture as shown in Figure 18. Specifically, an attention mechanism is described as the mapping from a query $\mathbf{q} \in \mathbb{R}^E$ and a set of key-value pairs, i.e., $\boldsymbol{K} \in \mathbb{R}^{M \times E}$ and $\boldsymbol{V} \in \mathbb{R}^{M \times E}$ ($M$ and $E$ are total number of options and embedding dimensions), to an output:

$$Attention(\mathbf{q}, \boldsymbol{K}, \boldsymbol{V}) = \text{softmax}(\frac{\mathbf{q}\boldsymbol{K}^T}{\sqrt{E}})\boldsymbol{V} \qquad (46)$$

A Multi-Head Attention MHA($\mathbf{q}, \boldsymbol{K}, \boldsymbol{V}$) is a linear projection of h (number of heads) concatenated linearly projected $Attention$ outputs:

$$\text{MHA}(\mathbf{q}, \boldsymbol{K}, \boldsymbol{V}) = \text{Concat}[\text{head}_1, \dots, \text{head}_h]\boldsymbol{W}^H \qquad (47)$$
$$\text{where head}_i = Attention(\mathbf{q}\boldsymbol{W}_i^q, \boldsymbol{K}\boldsymbol{W}_i^K, \boldsymbol{V}\boldsymbol{W}_i^V)$$

where projections are parameter matrices $\boldsymbol{W}_i^q \in \mathbb{R}^{E \times E}$, $\boldsymbol{W}_i^K \in \mathbb{R}^{E \times E}$, $\boldsymbol{W}_i^V \in \mathbb{R}^{E \times E}$, $\boldsymbol{W}_i^O \in \mathbb{R}^{hE \times E}$. In this paper we use MHA as one building block as illustrated in Figure 18.

We implement the *option policy* $P(\bar{\mathbf{o}}_t | \mathbf{s}_t, \bar{\mathbf{o}}_{t-1}; \boldsymbol{W_o})$ as the encoder and treat the option embedding matrix $\boldsymbol{W_o}$ as encoder's parameters. Specifically, we define the *option encoder* as:

$$\bar{\mathbf{o}}_t \sim \text{Categorical}(\text{Clustering}(\mathbf{s}_t, \bar{\mathbf{o}}_{t-1}, \boldsymbol{W_o})) \qquad (48)$$

where $Categorical(\cdot)$ is a $K$-dimensional categorical distribution, $K$ is the number of options, and distances between the pair $[\mathbf{s}_t, \hat{\mathbf{o}}_{t-1}]$ (where $\hat{\mathbf{o}}_{t-1} = \boldsymbol{W_o}\bar{\mathbf{o}}_{t-1}$).Under this configuration, options can be seen as clustering centroids. The problem of selecting the next option $\mathbf{o}_t$ is equivalent to calculate which clustering centroid $\hat{\mathbf{o}}$ in embedding matrix $\boldsymbol{W_o} = [\hat{\mathbf{o}}_1, ..., \hat{\mathbf{o}}_K]$ is closest to the projected state-option pair FFN($[\mathbf{s}_t, \hat{\mathbf{o}}_{t-1}]$) by an efficient MHA-based *clustering module*:

$$Clustering(\mathbf{s}_t, \bar{\mathbf{o}}_{t-1}, \boldsymbol{W_o}) = \text{FFN}(\text{MHA}(\text{Query} = \text{FFN}([\mathbf{s}_t, \hat{\mathbf{o}}_{t-1}]), \text{Key=Value} = \boldsymbol{W_o})) \quad (49)$$

The *action policy* can be simply implemented as one decoder, which learns to decode $\hat{\mathbf{o}}_t$ and $\mathbf{s}_t$ into primary actions $\mathbf{a}_t$.

$$\mathbf{a}_t \sim Gaussian(\text{FFN}([\mathbf{s}_t, \hat{\mathbf{o}}_t])) \qquad (50)$$

Because of the *Markovian option-value function* $\bar{V}[\mathbf{s}_{t+1}, \bar{\mathbf{o}}_t]$ is an expectation of the *option-value function* $Q_O[\mathbf{s}_{t+1}, \bar{\mathbf{o}}_{t+1}]$ in Eq. (6), we only need to model only one critic function: $Q_O = \text{FFN}(\mathbf{s}_t, \bar{\mathbf{o}}_t)$, where $Q_O$ is also a decoder of $\mathbf{s}_t$ and $\bar{\mathbf{o}}_t$. We summarize the detailed algorithm in Appendix F and upload our code in supplementary materials.

### G.2 HYPERPARAMETERS

In this section we summarize our implementation details. For a fair comparison, all baselines: DAC+PPO (Zhang & Whiteson, 2019), AHP+PPO (Levy & Shimkin, 2011), PPOC (Klissarov et al., 2017), OC (Bacon et al., 2017) and PPO (Schulman et al., 2017) are from DAC's open source Github repo: https://github.com/ShangtongZhang/DeepRL/tree/DAC. Hyper-parameters used in DAC (Zhang & Whiteson, 2019) for all these baselines are kept unchanged.

**MOPG Architecture:** For all experiments, our implementation of MOPG is exactly the same as Figure 18 (b). We use Pytorch to build neural networks. Specifically, for skill policy module, we use a skill context matrix $\boldsymbol{W}_S \in \mathbb{R}^{4 \times 40}$ which has 4 skills (4 rows) and an embedding size of 40 (40 columns). For Multi-Head Attention, we use Pytorch's built-in MultiheadAttention function[6] with $num\_heads = 1$ and $embed\_dim = 40$. For layer normalization we use Pytorch's built-in function LayerNorm [7]. For Feed Forward Networks (FNN), we use a 2 layer FNN with ReLu function as activation function with input size of 40, hidden size of 64, and output size of 64 neurons. For Linear layer, we use built-in Linear function[8] to map FFN's outputs to 4 dimension. Each dimension acts

---

[6]https://pytorch.org/docs/stable/generated/torch.nn.MultiheadAttention.html
[7]https://pytorch.org/docs/stable/generated/torch.nn.LayerNorm.html
[8]https://pytorch.org/docs/stable/generated/torch.nn.Linear.html

like a logit for each skill and is used as density in Categorical distribution[9]. For both action policy and critic module, FFNs are of the same size as the one used in the skill policy.

**Preprocessing:** States are normalized by a running estimation of mean and std.

**Hyperparameters of PPO:** For a fair comparison, we use exactly the same parameters of PPO as DAC. Specifically:

- Optimizer: Adam with $\epsilon = 10^{-5}$ and an initial learning rate $3 \times 10^{-4}$
- Discount ratio $\gamma$: 0.99
- GAE coefficient: 0.95
- Gradient clip by norm: 0.5
- Rollout length: 2048 environment steps
- Optimization epochs: 10
- Optimization batch size: 64
- Action probability ratio clip: 0.2

**Computing Infrastructure:** We conducted our experiments on an Intel® Core™ i9-9900X CPU @ 3.50GHz with a single thread and process with PyTorch.

---

[9]https://github.com/pytorch/pytorch/blob/master/torch/distributions/categorical.py

