# OpenReview forum: "HiT-MDP: Learning the SMDP option framework on MDPs with Hidden Temporal Embeddings"
_ICLR.cc/2023/Conference — ICLR 2023 poster_

### Official Review · Reviewer_CFgS · 2022-10-22

**Confidence:** 2
**Correctness:** 3
**Technical Novelty And Significance:** 3
**Empirical Novelty And Significance:** 3
**Recommendation:** 6

**Clarity, Quality, Novelty And Reproducibility:**

* Clarity: Mostly good.
* Quality and Novelty: The quality of this paper is pretty good and the proposed HiT-MDP is novel.
* Reproducibility: The source code is provided. I would assume the results are reproducible.

**Strength And Weaknesses:**

Strength:
* The writing is in general clear.
* The introduced HiT-MDP formulation is novel.

Questions:
* Can the authors provide more visualizations of the learned options?
* From Figure 6, it seems only 2 out of 4 options are useful. Is it because the environment is relatively simple? Does more complex environments (e.g. Humanoid) result in more options?
* Can the authors conduct similar analysis (as in Figure 5,6) for other environments, such as Humanoid or Hopper?
* How will the proposed method work in video games like Atari or Procgen? Can it learn some interesting options?

Minor issues:
* Please consider increasing the font size in Figure 1, 2, 5 and 6.

**Summary Of The Paper:**

This paper studies option learning in RL, and introduces a new framework (HiT-MDP). The authors prove that HiT-MDP is homomorphic equivalent to the standard SMDP formulation, and derive an on-policy policy gradient method with the new HiT-MDP formulation. On Mujuco environments, the proposed method is shown to be effective, exhibit smaller faster convergence and enjoy better interpretability.

**Summary Of The Review:**

In summary, this paper introduces a new HiT-MDP framework for option learning in RL. Based on HiT-MDP, the authors derive a policy gradient method and validate its effectiveness and advantages over exiting methods on Mujoco.

---

> ### Author Response · Authors · 2022-11-18
> **Response to Reviewer CFgS**
>
> We start by thanking the reviewer for his detailed feedback. We
> proceed to answer each of questions and thank you in advance for
> your patience in reading our elaborate reply. Due to space
> limitation, we have to leave all updates in the appendix for
> reviewer's reference (blue font).
>
> ### Q1: More visualization of the learned options? ###
>
> Answer:
>
> We have added the visualization of Humanoid-standup environment's
> learned options in Appendix B.2 (Figure 8 - 11, Page 15) and B.3
> (Figure 13, Page 17) for reviewer's reference.
>
>
> ### Q2: Task's Complexity v.s. Option switching frequency? ###
>
> Answer:
>
> We have added:
> 1. the visualization of Humanoid-standup environment's
> 2. option switching frequency with respect to the magnitude of
>    entropy in Appendix B (Figure 9-11) for reviewer's reference.
>
> The reviewer is correct that Environments' complexity affect
> usage of options and we especially appreciate the reviewer for
> picking this up. We only showed Halfcheetah result because this
> is kind of the standard environment to show and easy to
> understand. We surprisingly find that Humanoid-Standup task is
> also very interesting for demonstrating different option's
> functionalities. To demonstrate this we have added one more
> illustration of Humanoid-Standup task (Figure 10) in Appendix B.2
> for reviewer's reference.
>
> Besides task complexity, the usage of options is also determined
> by the magnitude of entropy coefficients. Generally speaking,
> agents tend to use more options with larger entropy since there
> are more randomness in the master policy. To illustrate this we
> added one more figure in Appendix B.2 (Figure 11).
>
> However, the magnitude of entropy coefficients is task specific.
> To demonstrate the effectiveness and robustness of our work, we
> did not tune hyperparameters in all reported performance and fix
> it to 0.01.
>
>
> ### Q3: Can the authors conduct similar analysis (as in Figure 5,6) for other environments, such as Humanoid or Hopper? ###
>
> Answer:
>
> For reviewer's reference, we have added Humanoid example in both
> Appendix B.2 (Figure 10, page 15) and provide a updated (Figure
> 13, page 17):
>
> 1. For Figure 5 (interpretation of option embedding vector), we
> interpret an embedding dimension which has effects to all actions
> to the same direction, and visualize the Halfcheetah's pose for
> the ease of understanding.
>
> 2. For Figure 6 (option switching frequencies and execution
> duration), we added the HumanoidStandupV2 simulation (Figure 9,10) and compared it with HalfcheetahV2 (Figure 8) results.
>
> We find that although all option baselines including our methods
> can find useful options in easy environment like HalfcheetahV2,
> this is not the case in harder environment like
> HumanoidStandupV2. As can be seen in Figure 9, In harder
> HumanoidStandupV2 task, all baselines fail to learn useful
> options, while Figure 10 shows that our method is able to learn
> disentangled options with clear options composition. This
> performance boost comes from both HiT-MDP's sample efficiency and
> MOPG's maximum entropy framework. Standard option frameworks have
> tendencies to learn either degenerate options [3] (short
> execution time) that switching back-and-forth too frequently, or
> dominant options [1] (long execution time) that executing through
> the whole episode. Both cases severely impair performance.
> Manually increase the option switch frequency is not necessarily
> increase an algorithm's performance.
>
> This is exactly our contribution and highlights our novelty: we
> enable to learn option frequency in an end-to-end manner. MOPG is
> a Maximum entropy algorithm and includes an
> information-theoretical intrinsic reward to encourage consecutive
> executions of options and entropy terms to encourage explorations
> of options. The whole algorithm can be solved in an end-to-end
> manner under the structional variational inference framework.
> Unlike [3] is a biased solution, we further theoretically proved
> that HiT-MDP is unbiased and converges to the optimal value
> (Markovian Option Policy Evaluation (Theorem 3.3), Improvement
> (Theorem 4.1) and Iteration Theorems (Theorem E.2)).
>
>
> ### Q4: How will the proposed method work in video games like Atari or Procgen? Can it learn some interesting options?
>
> Unfortunately, we do not have enough time for this update. We will
> include the Atari experiment in future version.

---

> ### Author Response · Authors · 2022-11-25
> **A Gentle Reminder**
>
> Dear Reviewer,
>
> We have revised our paper according to your advice. Please take a look and feel free to tell us if you have further concerns. We deeply appreciate your time and patience.
>
> Many thanks!

---

> > ### Comment · Reviewer_CFgS · 2022-11-30
> > **Thank you for the reply!**
> >
> > Thank the authors for the clarification and the additional visualizations. The response has addressed my concerns.

---

### Official Review · Reviewer_uUrN · 2022-10-24

**Confidence:** 4
**Correctness:** 4
**Technical Novelty And Significance:** 3
**Empirical Novelty And Significance:** 3
**Recommendation:** 8

**Clarity, Quality, Novelty And Reproducibility:**

Quality: The paper presents the first HIT-MDP and fully demonstrates the superiority of the framework over the traditional standard option
framework with rich and detailed experiments of high quality.

Clarity: This work follows the process of posing an existing problem, analyzing the problem, constructing a method to solve it, and performing a formula derivation to prove the effectiveness of the strategy. The exposition is well organized, the proof process is rigorous and scientific, and the derivation of formulas is detailed and convincing. The paper is illustrated with pictures where appropriate, and the premises and details of each formula are explained.

Novelty: The author's design is clever, the exposition is rigorous, and the content is original.


**Strength And Weaknesses:**

Pros:
The paper is very well motivated. The authors precisely present the shortcomings of the current standard option framework based on semi-Markovian decision processes and propose a new HIT-MDP that addresses the shortcomings of the original framework one by
one. It is also very clearly formulated. The arguments are clear, the derivation of formulas is rigorous. Even the lengthy appendix is fairly easy to parse, with straightforward proofs and relevant details. The most commendable thing is that the author has an innovative spirit. This paper is the first to propose an MDP equivalent to SMDP-Option, which provides reference and guidance for subsequent further research.

Cons:
The related work section is placed after the experimental section and before the conclusion and is somewhat abbreviated.


**Summary Of The Paper:**

This paper proposes a novel  Markov Decision Process (MDP),  namely, the Hidden Temporal MDP  (HiT-MDP), to solve the semi-Markov
decision process that is unstable, unoptimizable, and inefficient in sampling ,and proves that the option-induced HiT-MDP is homomorphic
equivalent to the option-induced SMDP. Then an efficient algorithm based on structured mutation reasoning is derived, which leads to a new
method of finding stable options under the maximum entropy reinforcement learning framework. Finally, HiT-MDP shows excellent performance in a wide range of configurations.

**Summary Of The Review:**

I think this is a good paper with solid theoretical analysis and contributions, the problems analyzed are very important, the proposed
architecture is feasible, the proof process is complete, and needs to be understood as deeply as this work.

---

> ### Author Response · Authors · 2022-11-18
> **Response to Reviewer uUrN**
>
> Dear Reviewer uUrN,
>
> Thank you for reviewing the paper! We are glad that you find our proposed theorems are well-motivated and the derivations are complete. We are very grateful that you appreciated our innovation!
>
> Due to the page limitation, we are sorry to only make small changes to the related works section. In this revision, we have some interesting visualizations included in the Appendix. We summarized below in case you are interested:
>
> We visualized all baselines' option switching frequencies in the easy (HalfCheetah) and hard (HumanoidStandupV2) environments (Appendix B.2, Figure 8-10). We find that:
>
> - In the easy environment HalfCheetah, all baselines including our method are able to discover useful options. There is no big difference between our method and other baselines in terms of switching frequency. Given that our method doubled the performance in HalfCheetah, it proves that the proposed HiT-MDP could be more sample efficient than the others in easy environments.
>
> - In the hard environment HumanoidStandupV2, we add a visualization (Figure 10) of option compositions and typical pose of the humanoid for each option. It appears that baseline methods tend to suffer from either degenerate (very short execution duration) options or dominant options (1 option dominates most timesteps). On the contrary, our method is able to discover disentangled options and a clear composition schedule. This advantage comes from the maximum entropy nature of the MOPG algorithm which is able to learn diversified options with coherent intrinsic-reward in an end-to-end manner.

---

> > ### Comment · Reviewer_uUrN · 2022-11-18
> > **Response to author reply**
> >
> > Thanks a million for clarifying my previous concerns. As I mentioned earlier, overall this is a good paper with solid theoretical analysis and contributions. The empirical results are also strong, especially with the newly added discussions about option switching frequencies. Thus, I maintain my original rating and recommend to accept this paper.

---

### Official Review · Reviewer_ySTp · 2022-10-25

**Confidence:** 3
**Clarity, Quality, Novelty And Reproducibility:** OK writing.
**Correctness:** 4
**Technical Novelty And Significance:** 1
**Empirical Novelty And Significance:** Not applicable
**Recommendation:** 3

**Strength And Weaknesses:**

Hard to learn from the survey. No applications discussed.

Strength

The submission is well motivated, focusing on two key challenges in option RL.

A novel formulation of option RL is proposed, which facilitates new option RL methods.

The results provided by the submission are thorough and complete, with both theoretical and empirical support for the methodology.

Weaknesses

It is not very clear to me why the proposed new formulation and methodology can have a better sample complexity than existing ones. At the beginning, the authors state that the reason for the poor sample complexity of option RL is that one option lasts for multiple steps, which wastes observations. However, this is also true under the proposed HiT-MDP formulation: in each time step, there is a high probability that the option stays the same as the previous step. Both formulations have strong probabilities to maintain the selected option for multiple steps. So, why is the proposed one more sample efficient?

Would it be possible for authors to provide the option change frequency of each method in experiments? In experiments, the proposed method shows better sample complexity than existing ones. I am wondering whether this is because the proposed method changes options more frequently than the competing ones? If so, can we just manually increase the option switch frequency of existing methods to achieve similar sample complexity gain?


**Summary Of The Paper:**

Mostly survey, not clear what the contribution is. The submission considers option RL. The major challenges the submission tackles is (1) the poor sample efficiency and (2) hard optimization for such a problem. The authors argue that the cause of the two challenges is the semi-MDP framework of existing option RL methods. Therefore, to deal with this issue, the submission reformulates the option RL problem into an HMM. Therefore, RL methods for HMM can be leveraged to deal with the two challenges in option RL.

Both theoretical and empirical results are provided supporting the proposed method.


**Summary Of The Review:**

It's a survey paper with citations permeating through sections 6 and 7.
The submission proposes both novel formulation and novel methodology for option RL. Both theoretical and synthetic results are provided supporting the proposed method. However, I am not quite sure why the proposed method and formulation are able to provide such performance improvement.

---

> ### Author Response · Authors · 2022-11-18
> **Response to Reviewer ySTp**
>
> We start by thanking the reviewer for his detailed feedback. We
> proceed to answer each of questions and thank you in advance for
> your patience in reading our elaborate reply.
>
>
> ### Q1: Why HiT-MDP is more sample efficient than SMDP-based option framewoks? ###
>
> Answer:
>
> We understand the reviewer's confusion because the difference is
> in the training stage not inference stage (Explanation below can
> also be seen in DAC[1] and IOPG[2]). In standard SMDP-based
> option frameworks, since an option executes multiple steps, an
> update (i.e., one SGD iteration) of the policy-gradient based
> algorithms can only happen when the option terminates. This means
> that standard SMDP-based policy-gradient algorithms for option
> frameworks consumes *multiple timesteps of samples* for only 1
> update.
>
> On the contrary, MDP-based algorithms samples from the master
> policy at every step with no addtional computational cost,
> therefore, one update of MOPG only consumes *1 timestep sample*
> for each update. This means MDP-based algorithms can update
> policies at every timestep and thus is far more efficient than
> SMDP-based ones during the training stage.
>
> Therefore, MDP-based algorithms converges significantly faster
> than standard SMDP-based algorithms given the same amount of
> samples, in another word, is more sample efficient.
>
> Although there are many variants trying to solve sample
> complexity issues under this direction, as reviewer uUrN pointed
> out that this is the first work proposing an MDP equivalent
> option framework with rigorous theoretically proof, and is
> impactful to the option community. We hope our explanation can
> help reviewer better evaluate our novelty.
>
>
> ### Q2. Provide the option change frequency of each method in experiments ###
>
> Answer:
>
> We have added option frequency figures for all methods in
> Appendix B.2 for reviewer's reference.
>
> Option switching frequency is double-edged in terms of
> performance. The option switching frequency is not monotonically
> related to the performance. As can be seen in Figure 8, in simple
> environment like Halfcheetah, there are not much switching
> frequency differences between option variants, including our
> method. Therefore, the performance boost mainly comes from
> HiT-MDPs' sample efficiency.
>
> However, this is not the case in harder environment. As can be
> seen in Figure 8 and 9. In harder HumanoidStandupV2 task, all
> baselines fail to learn useful options, while our method is able
> to learn disentangled options with clear options composition.
> This performance boost comes from both HiT-MDP's sample
> efficiency and MOPG's maximum entropy framework. Standard option
> frameworks have tendencies to learn either degenerate options [3]
> (short execution time) that switching back-and-forth too
> frequently, or dominant options [1] (long execution time) that
> executing through the whole episode. Both cases severely impair
> performance. Manually increasing the option switch frequency is not
> necessarily increase an algorithm's performance.
>
> This is exactly our contribution and highlights our novelty: we
> enable to learn option frequency in an end-to-end manner. MOPG is
> a Maximum entropy algorithm and includes an
> information-theoretical intrinsic reward to encourage consecutive
> executions of options and entropy terms to encourage explorations
> of options. The whole algorithm can be solved in an end-to-end
> manner under the structional variational inference framework.
> Unlike [3] is a biased solution, we further theoretically proved
> that HiT-MDP is unbiased and converges to the optimal value.
>
>
> ----------------------
>
> At last we would like to thank the reviewer for your time and
> highlight our contribution.
>
> In this work we propose a very important theorem: the SMDP-based
> option framework has an MDP equivalence. As pointed out by
> reviewer uUrN, we believe that this equivalence theorem is a very
> important finding for both option community and probably the
> broader Hierarchical Reinforcement Learning community.
>
> One of the most important application is discussed in Section
> 5.3. The equivalence theorem enables the standard tuple option to
> be represented by a simple option embedding vector. It unlocks
> the doors for the option framework to employ a large-scale
> foundation model such as Transformer to learn temporal
> embeddings. Due to page limitation we have to defer this research
> to later stage. But still we hope our explanation can help the
> reviewer better evaluate our contribution.
>
>
> [1]: Shangtong Zhang and Shimon Whiteson. DAC: The double actor-critic architecture for learning options. In Advances in Neural Information Processing Systems, pp. 2,012–2,022, 2019.
>
> [2]: Matthew Smith. An inference-based policy gradient method for learning options. In International Conference on Machine Learning, pp. 4,703–4,712, 2018.
>
> [3]: Jean Harb, Pierre-Luc Bacon, Martin Klissarov, and Doina Precup. When waiting is not an option: Learning options with a deliberation cost. In Thirty-Second AAAI Conference on Artificial Intelligence, 2018.

---

> ### Author Response · Authors · 2022-12-05
> **A Gentle Reminder**
>
> Dear Reviewer,
>
> We have revised our paper according to your advice. Please take a look and feel free to tell us if you have further concerns. We deeply appreciate your time and patience.
>
> Many thanks!

---

### Official Review · Reviewer_iC2w · 2022-10-31

**Confidence:** 3
**Correctness:** 3
**Technical Novelty And Significance:** 2
**Empirical Novelty And Significance:** 2
**Recommendation:** 5

**Clarity, Quality, Novelty And Reproducibility:**

See questions above. Implementation details are shared but the code is not released. Typo in p6: dynamix -> dynamics

**Strength And Weaknesses:**

$\textbf{Strength:}$
The authors propose a novel MDP view of SMDP that models the options as part of the states and actions, leading to the definition of several option-value functions. The proposed MDP model can potentially extend various existing RL algorithms that solve MDP to the SMDP problem setup.

$\textbf{Weaknesses and Questions:}$

- How does MOPG compare with Attention Option-Critic [1], which also uses attention network to encode option?


- The theory is not sufficient for me. There is no proof or discussion on the convergence or sample efficiency of MOPG. The equivalence between SMDP and MDP could be more significant if authors can provide such type of analyses.

- The contribution is limited, as the proposed MOPG still falls in the well-established PPO framework, and similar architecture was explored before in [1].

[1] Chunduru and Precup. Attention Option-Critic. (2022).


**Summary Of The Paper:**

This paper aims to solve Semi-Markov decision processes (SMDP) by reformulating the SMDP into an ordinary MDP. In particular, the authors propose a novel MDP view of SMDP that models the options as part of the states and actions, leading to the definition of several option-value functions. The authors propose to utilize an energy-based parameterization in the transition model, which results in an entropy regularizer term in policy optimization. The authors further solve such policy optimization problems by policy gradient and propose the MOPG algorithm for solving SMDPs. Finally, the authors conduct several experiments on the MuJuCo environment and compare MOPG with several SOTA algorithms.

**Summary Of The Review:**

The idea is interesting, but the contribution is marginal to me.

---

> ### Author Response · Authors · 2022-11-18
> **Response to Reviewer iC2w**
>
> We start by thanking the reviewer for his detailed feedback. We
> proceed to answer each of questions and thank you in advance for
> your patience in reading our elaborate reply. In this version we
> made following updates (blue font):
>
> ### Q1 & Q3: How does MOPG compare with Attention Option-Critic?
>
> Answer:
>
> We duly cited Attention Option-Critic (AOC). (2022) as the first
> work to introduce attention mechanism into option framewoks in
> Section 5.3, Section 6 (Related Works) and Appendix G.
>
> As we explained in the Appendix G, AOC and our method contributes
> to the option framework from two very different aspects: in AOC,
> an option is a tuple of $(I, \beta, \pi_A)$ while in our
> implementation an option is an option embedding vector $\hat{o}$.
> This is because the attention mechanism in AOC is employed as a
> mask over *the observation space* $\ddot{s} = \ddot{W}_o \odot
> s$. In our case, the attention mechanism is used as a distance
> measure over *the action space* (eq. 46 and eq. 47) for learning
> option embeddings: o_t ∼ Categorical(Clustering(s_t , o_{t−1},
> W_o)).
>
> Also our main contribution in this work is theoretically prove
> the existence of an MDP equivalence (HiT-MDP) to the SMDP based
> option framework. Therefore, in experiments (Section 5.1 and 5.2)
> we focus on comparing HiT-MDPs' performance to other works (AHP,
> IOPG and DAC) which also modify the option framework's underlying
> decision process. The attention mechanism proposed in Section 5.3
> is only an implementation choice of function approximation for
> the option policy and can be replaced by other distance measures.
>
>
> ### Q2 and Q3: Proof of the convergence and sample efficiency of MOPG? The novelty of MOPG compares to PPO?
>
> Answer:
>
> For MOPG's convergence proof, we thank the reviewer for pointing
> this out. We missed the policy iteration theorem in our last
> version and have added the The Markovian Option Policy Iteration
> Theorem in Appendix E.1 in this version.
>
> We would like to mention that the convergence of HiT-MDPs was
> proved by Theorem 3.3.
>
> It is also worth to mention that the significant of the
> equivalence between HiT-MDP and option-induced SMDP is not
> affected by the proof of MOPG:
>
> 1. Proposing HiT-MDPs and theoretically prove the equivalence to
>    the SMDP based option frameworks. As mentioned by reviewer
>    uUrN, proof of the equivalence is rigorous. We both proved the
>    homomorphic equivalence between two Decision processes and the
>    convergence of the HiT-MDP to the conventional value function.
>    Therefore, the proof of this equivalence is complete.
> 2. MOPG is a policy-gradients based algorithm for solving
>    HiT-MDPs. It's convergence rate and sample complexity do not
>    affect the equivalence. With the ELBO derived in Section 4,
>    HiT-MDPs can also incorporates other algorithms (such as
>    SAC-like off-policy algorithms).
>
>
> Comparison to PPO:
>
> As for the novelty of MOPG compared to PPO, these two algorithms
> contributes to solving HiT-MDPs at different level: MOPG is a
> policy-gradient based algorithm, the main contribution of MOPG is
> deriving the option and action gradient theorems in Section 4.
> PPO is only an approximation algorithm for safely updating policy
> gradients in an approximated trusted region. PPO can be used to
> safely learn MOPG's gradients but itself does not tell MOPG's
> gradient function (Theorem 4.2 and 4.3). From this point of view,
> MOPG and DAC (which also employs PPO for safe learning) is at the
> same level.
>
> From the performance aspect, we compared our method with PPOC and
> DAC+PPO in Figure 4. As experiment shows, MOPG also has a
> significant performance boost over PPOC and DAC+PPO in infinite
> horizon environments and transfer learning settings. This further
> proof that MOPG's performance boost is not come from PPO.

---

> ### Author Response · Authors · 2022-12-05
> **A Gentle Reminder**
>
> Dear Reviewer,
>
> We have revised our paper according to your advice. Please take a look and feel free to tell us if you have further concerns. We deeply appreciate your time and patience.
>
> Many thanks!

---

### Decision · Program_Chairs · 2023-01-20

**Decision:**

Accept: poster

**Justification For Why Not Higher Score:**

While this is a solid paper, I believe the topic is somewhat niche to warrant broader dissemination at the conference.

**Justification For Why Not Lower Score:**

Strengths outweigh the weaknesses of this paper. Overall, a solid contribution.

**Metareview: Summary, Strengths And Weaknesses:**

I thank the authors for their submission and active engagement during the discussion period. This is a borderline paper. On the positive side, reviewers acknowledged the novel MDP formulation accounting for options [iC2w]. They found the work well motivated and focusing on important problems in option learning [ySTp]. The method is novel [ySTp,uUrN] and the empirical investigation thorough [ySTp]. On the negative side, there are concerns around the method's sample efficiency [ySTp] and comparison to prior work [iC2w]. I believe the authors have addressed reviewer's iC2w concerns appropriately. Likewise, I believe the rebuttal addresses ySTp's concerns regarding the option frequency. Therefore, I recommend acceptance but strongly encourage the authors to take the reviewer feedback into account for the camera ready version of the paper.

**Note From Pc:**

if the above contains the word "oral" or "spotlight" please see: "oral" presentation means -> notable-top-5% and "spotlight" means -> notable-top-25%. As stated in our emails, we are disassociating presentation type from AC recommendations